# PRACTICAL APPROACHES FOR FAIR LEARNING WITH MULTITYPE AND MULTIVARIATE SENSITIVE ATTRIBUTES

## ABSTRACT

It is important to guarantee that machine learning algorithms deployed in the real world do not result in unfairness or unintended social consequences. Fair ML has largely focused on the protection of single attributes in the simpler setting where both attributes and target outcomes are binary. However, the practical application in many a real-world problem entails the simultaneous protection of multiple sensitive attributes, which are often not simply binary, but continuous or categorical. To address this more challenging task, we introduce FairCOCCO, a fairness measure built on cross-covariance operators on reproducing kernel Hilbert Spaces. This leads to two practical tools: first, the FairCOCCO Score, a normalized metric that can quantify fairness in settings with single or *multiple* sensitive attributes of *arbitrary* type; and second, a subsequent regularization term that can be incorporated into arbitrary learning objectives to obtain fair predictors. These contributions address crucial gaps in the algorithmic fairness literature, and we empirically demonstrate consistent improvements against state-of-the-art techniques in balancing predictive power and fairness on real-world datasets.

## 1 INTRODUCTION

There is a clear need for scalable and practical methods that can be easily incorporated into machine learning (ML) operations, in order to make sure they don't inadvertently disadvantage one group over another. The ML community has responded with a number of methods designed to ensure that predictive models are *fair* (under a variety of definitions that we shall explore later) (Caton & Haas, 2020). Perhaps due to the archetypal fairness example, an investigation into the COMPAS software that found racial discrimination in the assessment of risk of recidivism (Angwin et al., 2016), most of the focus has been on *single, binary* variables - in this case *race* being treated as an indicator of whether an individual was *black* or *white*. This, combined with a discrete target, allows for easy analysis of fairness criteria such as demographic parity and equalized odds (Barocas & Selbst, 2016; Hardt et al., 2016), through the rates of outcomes in the confusion matrix of the subgroups.

The problem is, however, that in many practical applications we may have multiple attributes which we would like to protect, for example both *race* and *sex* - indeed U.S. federal law protects groups from discrimination based on nine protected classes (EEOC, 2021). Algorithms deployed in the real-world therefore need to be capable of protecting multiple attributes both *jointly* (e.g. 'black woman') and *individually* (e.g. 'black' and 'woman'). This is non-trivial and cannot be simply achieved by introducing separate fairness conditions for each attribute. Such an approach both does not provide joint protection of sensitive attributes and complicates matters by introducing additional hyperparameters that need to be traded-off against each other. Matters are further complicated by the fact that many sensitive attributes (e.g. age) and outcomes (e.g. credit limit) take on *continuous* values, for which calculated rates do not make sense. Existing methods simply discretise these into categorical bins, which leads to several issues in practice, as it entails thresholding and data sparsity effects while discarding element order information. As we shall see later in Section 4, this approach is unlikely to be optimal in delivering discriminative yet fair predictors.

**Contributions and Outline.** Consequently, we introduce two practical tools to the community, which we hope can be used to more easily incorporate fairness into a standard ML pipeline: a **(1) Fairness**

**metric.** We introduce the `FairCOCCO score`, a flexible normalized metric that can quantify the level of independence-based fairness in tasks with multitype and multivariate sensitive attributes by employing the cross-covariance operator on reproducing kernel Hilbert Spaces (RKHS); and a **(2) Fairness regulariser.** Based on the `FairCOCCO score`, we construct a fairness regulariser that can be easily added to arbitrary learning objectives for fairness-aware learning.

In what follows, we introduce current notions of fairness alongside contemporary methods to ensure fair learning (**Section 2**), before introducing our contributions and explain how they plug the crucial gaps in the literature (**Section 3**). With that established, we illustrate the practical advantages of `FairCOCCO` in a series of demonstrations on multiple real-world datasets across a variety of modalities, quantitatively demonstrating consistent improvements over state-of-the-art techniques (**Section 4**). We conclude with a discussion on future work and societal implications (**Section 5**).

## 2 BACKGROUND

**Fairness Notions** Let $d_X$, $d_Y$, $d_A$ be dimensions of measurable space $\mathcal{X} \subset \mathbb{R}^{d_X}$, $\mathcal{Y} \subset \mathbb{R}^{d_Y}$ and $\mathcal{A} \subset \mathbb{R}^{d_A}$, respectively. We introduce random variable $X$ defined on $\mathcal{X}$ to denote the features; $Y$ and $A$ are similarly defined and denote the target and sensitive attribute(s) that we want to protect (e.g. gender or race). Note that $A$ can be part of $X$, i.e. with a slight abuse of notation, we can write $A \subset X$.

We are mainly concerned with quantifying *group fairness*, which requires that protected groups (e.g. black applicants) be treated similarly to advantaged groups (e.g. white applicants) (Caton & Haas, 2020). In Table 1, we highlight four popular

Table 1: **Popular definitions of fairness.** Defined in terms of (conditional) independence requirements.

| Definition | Requirement |
|:---:|:---:|
| FTU | $A \perp\!\!\!\perp (\hat{Y} \mid X \setminus A)$ |
| DP | $A \perp\!\!\!\perp \hat{Y}$ |
| EO | $(A \perp\!\!\!\perp \hat{Y}) \mid Y$ |
| CAL | $(A \perp\!\!\!\perp Y) \mid \hat{Y}$ |

definitions and how each quantifies a different aspect of fairness. *Fairness through unawareness* (FTU) (Grgic-Hlaca et al., 2016) prohibits the algorithm from using sensitive attributes explicitly in making predictions. While straightforward to implement, this method ignores the indirect discriminatory effect of proxy covariates that are correlated with $A$, e.g. "redlining" (Avery et al., 2009). *Demographic parity* (DP) (Barocas & Selbst, 2016; Zafar et al., 2017) accounts for indirect discrimination, by requiring statistical independence between predictions and attributes $\hat{Y} \perp\!\!\!\perp A$. Evidently, this strict notion sacrifices predictive utility by ignoring all correlations between $Y$ and $A$, thereby precluding the optimal predictor. Dwork et al. (2012), most notably, argues that this approach permits laziness, which can hurt fairness in the long run. To address some of these concerns, Hardt et al. (2016) introduced *equalized odds* (EO), requiring that predictions $\hat{Y}$ and attributes $A$ are independent given the true outcome $Y$, i.e. $\hat{Y} \perp\!\!\!\perp A \mid Y$. This approach recognizes that sensitive attributes have predictive value, but only allows $A$ to influence $\hat{Y}$ to the extent allowed for by the true outcome $Y$. For binary predictions and sensitive attributes, a metric known as *difference in equal opportunity* (DEO) highlights the different predictions made based on different group memberships:

$$DEO = |P(\hat{Y}|A = 1, Y = 1) - P(\hat{Y}|A = 0, Y = 1)|$$

Additional notions of fairness include *calibration* (CAL) (Kleinberg et al., 2016), which ensures that predictions are calibrated between subgroups, i.e. $Y \perp\!\!\!\perp A \mid \hat{Y}$. For a comprehensive review of fairness notions, we defer to §3 in Caton & Haas (2020). In the remaining sections, we illustrate our proposed methods using the framework of EO, but this is without loss of generality, as our method is compatible with any dependency-based fairness measure. It is important to note that there is no universal measure of fairness, and the correct notion depends on ethical, legal and technical contexts.

### 2.1 RELATED WORKS

Technical approaches to algorithmic fairness can be categorized into three main types: prior to modelling (pre-processing), during modelling (in-processing) or after modelling (post-processing) (del Barrio et al., 2020). The work herein falls into the category of *in-processing* techniques, which achieve fairness by incorporating either constraints or regularisers. Table 1 makes explicit the connection between fairness notions and (conditional) dependence. At the core of many algorithmic fairness techniques is how fairness is estimated and constrained. Much of the literature focuses on

Table 2: **Overview of related work for fairness-aware learning.** Comparison made on method of **fairness estimation**, **underlying model class** and the following desiderata: **(1)** supports continuous outcomes; **(2)** continuous attributes; **(3)** protects multiple attributes; **(4)** compatible with all dependency-based notions of fairness (as in Table 1).

| Method | Fairness Estimation | Predictive Model | (1) | (2) | (3) | (4) |
|---|---|---|---|---|---|---|
| Zemel et al. (2013) | Mutual information | Linear | ✗ | ✗ | ✗ | ✓ |
| Zafar et al. (2017) | Conditional covariance | Linear/Kernel | ✗ | ✗ | ✗ | ✗ |
| Donini et al. (2018) | Linear loss | Linear/Kernel | ✗ | ✗ | ✗ | ✗ |
| Cho et al. (2020) | Linear loss | Any | ✗ | ✗ | ✗ | ✓ |
| Mary et al. (2019) | Rényi correlation | Any | ✓ | ✓ | ✗ | ✓ |
| Steinberg et al. (2020b) | Mutual information | Any | ✗ | ✓ | ✗ | ✓ |
| Pérez-Suay et al. (2017) | Kernel measure | Linear/Kernel | ✓ | ✓ | ✓ | ✗ |
| FairCOCCO | Kernel measure | Any | ✓ | ✓ | ✓ | ✓ |

settings with a single, binary label and attribute (Kamishima et al., 2012; Goel et al., 2018; Jiang et al., 2020; Donini et al., 2018), where fairness quantification is straightforward by comparing rates of outcomes between subgroups. However, settings involving continuous variables are significantly more challenging (Bergsma, 2004). Recent efforts in fair regression (where only outcomes are continuous) (Agarwal et al., 2019; Chzhen et al., 2020) discretise continuous variables, but such approaches introduce unwanted threshold effects, discards order information and requires sufficient sample coverage in each bin.

**Protecting continuous attributes.** To pursue a fully continuous treatment, recent methods have made parametric or other assumptions to simplify conditional dependence criteria. Calders et al. (2013), Johnson et al. (2016b) and Bechavod & Ligett (2017) reduce the task of dependence minimization to minimizing the distances between moments of distributions. Donini et al. (2018) generalizes this to minimizing the distance between first moments of functions. Woodworth et al. (2017) and Zafar et al. (2017) similarly employ second moment relaxation to regularize only conditional covariance, corresponding to removing linear correlations only. Kamishima et al. (2012) introduced a first moment relaxation of mutual information (MI). However, such approaches are limiting as *weak* fairness measures that cannot fully capture important fairness effects and potentially lead to harm if the distribution assumptions are miss-specified (Daudin, 1980).[1]

Ideally, we hope for a *strong* measure that can accurately identify the level of fairness. Key approaches include kernel methods, MI, maximal correlation. Cho et al. (2020) employ kernel density estimation (KDE) to compute MI to enforce fairness. Lowy et al. (2021) and Mary et al. (2019) developed regularization using maximal correlation, but similarly rely on KDE, which does not scale to higher dimensions. Steinberg et al. (2020a) and Steinberg et al. (2020b) adopts a MI-based measure using density ratio estimation, but requires the training of an inner-loop probabilistic classifier.

**Protecting multiple attributes.** Few existing methods support protection of multiple attributes, even though this is a common and necessary requirement in practice. Kearns et al. (2018) highlighted *fairness gerrymandering*, in which a predictor appears to be fair on each individual attribute (e.g. black) but badly violates fairness when considering multiple sensitive attributes (e.g. black woman). Put formally, the prediction should be jointly independent (i.e. fair) to multiple sensitive attributes while also being independent to each individual attribute. Fortunately, this is already implied due to the *decomposition* property:

$$\hat{Y} \perp\!\!\!\perp (A_1, \ldots, A_{d_A}) \mid Y \; \Rightarrow \; \hat{Y} \perp\!\!\!\perp A_i \mid Y \;\; \forall \;\; i \in \{1, \ldots, d_A\} \tag{1}$$

However, we cannot naively extend existing methods to protect multiple attributes by introducing separate conditions on each attribute. This is evident, as the inverse proposition of (1) does not hold in general. In other words, while this naive approach can ensure individual protection, it does not guarantee protection of all attributes simultaneously. A related stream of research investigates *intersectional fairness* (Kearns et al., 2018; Foulds et al., 2020), which models combinatorial intersection of various subgroups. However, this only considers discrete attributes and outcomes, and one notion of fairness (DP). Table 2 provides an overview and comparison of related works.

---

[1] *Weak* and *strong*, as defined by (Daudin, 1980), refer to the strength of characterizations of dependence.

## 3   EVALUATING AND LEARNING FAIRNESS

In this section, we introduce `FairCOCCO`, a strong fairness measure from which we develop a metric and regulariser for fair learning. It applies kernel measures to quantify and control the level of dependence between algorithm predictions and protected attributes, such that the fairness requirements in Table 1 hold.

### 3.1   KERNEL MEASURE OF FAIRNESS

**Setup.** Let $\mathcal{H}_\mathcal{Y}$ denote the RKHS on $\mathcal{Y}$, with positive definite kernel $k_\mathcal{Y}$. $k_\mathcal{A}$ and $\mathcal{H}_\mathcal{A}$ are defined similarly. [2] Formally, the problem of interest is quantifying the conditional fairness between $\hat{Y}$ and $A$ given $Y$ on finite samples.

We propose a measure based on the conditional cross-covariance operator in Reproducing Kernel Hilbert Space (RKHS). A RKHS $\mathcal{H}_\mathcal{Y}$ is a Hilbert space of functions, in which each point evaluation $f(y)$, for any $y \in \mathcal{Y}$ and $f \in \mathcal{H}_\mathcal{Y}$, is a bounded linear functional. Distributions of variables can be embedded into the RKHS through kernels, where inference of higher order moments and dependence between distributions can be performed (Bach & Jordan, 2002; Gretton et al., 2005).

**Unconditional fairness.** We start by describing how operators in the RKHS can be used to evaluate fairness in the unconditional case (DP), by quantifying reliance of model predictions $\hat{Y}$ on sensitive attributes $A$. The cross-covariance operator (CCO) $\Sigma_{\hat{Y}A} : \mathcal{H}_\mathcal{A} \rightarrow \mathcal{H}_\mathcal{Y}$ is the unique, bounded operator that satisfies the relation:

$$\langle g, \Sigma_{\hat{Y}A} f \rangle_{\mathcal{H}_\mathcal{Y}} = \mathbb{E}_{\hat{Y}A}[f(\hat{Y})g(A)] - \mathbb{E}_{\hat{Y}}[f(\hat{Y})]\mathbb{E}_A[g(A)], \tag{2}$$

for all $f \in \mathcal{H}_\mathcal{Y}$ and $g \in \mathcal{H}_\mathcal{A}$. Intuitively, the $\Sigma_{\hat{Y}A}$ operator extends the covariance matrix defined on Euclidean spaces to represent higher (possibly infinite) order covariance between $\hat{Y}$ and $A$ through kernel mappings $f(X)$ and $g(Y)$. Additionally, we can obtain a normalized operator, i.e. the normalized cross-covariance operator (NOCCO) $V_{\hat{Y}A}$ (Baker, 1973):

$$V_{\hat{Y}A} = \Sigma_{\hat{Y}\hat{Y}}^{-\frac{1}{2}} \Sigma_{\hat{Y}A} \Sigma_{AA}^{-\frac{1}{2}}, \tag{3}$$

where $\Sigma_{\hat{Y}\hat{Y}}, \Sigma_{AA}$ are defined similarly to (2). This normalization is analogous to the relationship between covariance and correlation, and disentangles the influence of marginals while retaining the same dependence information. Intuitively, we have obtained a strong measure of correlation between sensitive attributes and fairness by leveraging the RKHS to represent higher-order moments.

**Conditional fairness.** For many notions of fairness (i.e. EO and CAL), we also require a measure of *conditional* fairness. We will frame the discussion around EO, where the prediction should be independent of the sensitive attribute given the true outcome $\hat{Y} \perp\!\!\!\perp A \,|\, Y$. It is straightforward to adapt this for CAL by swapping variables around. We can derive a normalized, conditional cross-covariance operator, by manipulating (3), i.e. $V_{\hat{Y}A|Y}$ (COCCO):

$$V_{\hat{Y}A|Y} = V_{\hat{Y}A} - V_{\hat{Y}Y}V_{YA} \tag{4}$$

where $V_{\hat{Y}Y}, V_{YA}$ are defined similarly to (3). In line with the intuition established previously, this operator measures higher-order partial correlation through function transformations $f(A) \,\forall\, f \in \mathcal{H}_\mathcal{A}$ and $g(\hat{Y}), h(Y) \,\forall\, g, \, h \in \mathcal{H}_\mathcal{Y}$. We round up this discussion by characterizing the relation between the $V_{\hat{Y}A|Y}$ operator and conditional fairness.

**Lemma 3.1 (COCCO and Conditional Fairness (Fukumizu et al., 2007))**
*Denote $\ddot{A} \triangleq (A, Y)$, and the product of kernels $k_{\ddot{A}} \triangleq k_\mathcal{A}k_\mathcal{Y}$, and further assuming $k_{\ddot{A}}$ is a characteristic kernel. Then:*

$$V_{\hat{Y}\ddot{A}|Y} = 0 \iff \hat{Y} \perp\!\!\!\perp A \,|\, Y \tag{5}$$

Note that $\ddot{A}$ denotes the extended variable set. For ease of notation, we write $V_{\hat{Y}A|Y}$ in place of $V_{\hat{Y}\ddot{A}|Y}$ from this point onward. (3) and (4) gives us a way to measure unconditional and conditional fairness, respectively, and lower values will indicate higher levels of fairness. Additionally, we note that (3) can be viewed as a special case of (4), where $\mathcal{Y} = \emptyset$, i.e. $V_{\hat{Y}A} = 0 \leftrightarrow \hat{Y} \perp\!\!\!\perp A$.

---

[2] We make mild assumptions on the involved RKHSs, assuming they are separable and square integrable (Gretton et al., 2005); and employ characteristic kernels, e.g. Gaussian and Laplacian kernels.

## 3.2 METRIC: FAIRCOCCO SCORE

Having described a kernel-based measure of fairness, we propose a fairness metric that is applicable to conditional and unconditional fairness as well as settings with multiple sensitive attributes of arbitrary (continuous or discrete) type. Many metrics (e.g. DEO (Hardt et al., 2016) to evaluate EO, and DI (Feldman et al., 2015) to evaluate DP) have been proposed for binary fairness settings. However, their utility is limited to classification tasks with single binary sensitive attributes. This is insufficient in real-world conditions, where there often exists many sensitive attributes that can be discrete or continuous. To address these challenges, we propose `FairCOCCO Score` that can evaluate fairness of several attributes of mixed type and for both continuous and discrete outcomes.

We start by summarizing the information contained in $V_{\hat{Y}A}$ into a single statistic using the squared Hilbert-Schmidt (HS) norm (Bach & Jordan, 2002):

$$I = ||V_{\hat{Y}A}||^2_{HS} \qquad (6)$$

This scalar value can be estimated from samples analytically, and we provide the complete closed-form expression in Appendix A. By Lemma 3.1, we know that $||V_{\hat{Y}A}||^2_{HS} = 0 \iff \hat{Y} \perp\!\!\!\perp A$. Thus, values closer to zero indicate higher levels of conditional fairness. However, while (6) is non-negative, it can be arbitrarily large. This makes it hard to interpret and compare across different tasks. To address this, we propose the normalized metric `FairCOCCO score`:

**Definition 3.2 (FairCOCCO Score)**

$$FairCOCCO \ Score \ (unconditional) = \frac{||V_{\hat{Y}A}||^2_{HS}}{||V_{\hat{Y}\hat{Y}}||_{HS}||V_{AA}||_{HS}} \qquad (7)$$

$$FairCOCCO \ Score \ (conditional) = \frac{||V_{\hat{Y}\ddot{A}\,|\,Y}||^2_{HS}}{||V_{\hat{Y}\hat{Y}|Y}||_{HS}||V_{\ddot{A}\ddot{A}|Y}||_{HS}} \qquad (8)$$

*which takes values in $[0, 1]$, where value closer to $0$ indicates higher levels of fairness, and vice versa.*

This normalization scheme is derived from the Cauchy-Schwarz inequality and can be understood as taking into account the (conditional) variance within each variable (c.f. relationship between covariance and correlation). In Appendix A, we derive the metric and its conditional counterpart and additionally demonstrate how the measure (6) can be used to perform (conditional) independence testing for additional transparency and interpretability.

The `FairCOCCO Score` can be used to measure any of the independence based notions of fairness. In particular, to make the connection with the Table 1 clear, the terms for the different notions can be expressed as:

$$I_{EO} = ||V_{\hat{Y}A|Y}||^2_{HS}, \qquad I_{CAL} = ||V_{YA|\hat{Y}}||^2_{HS}, \qquad I_{DP} = ||V_{\hat{Y}A}||^2_{HS} \qquad (9)$$

## 3.3 LEARNING: FAIRCOCCO LEARNING

Now that we have established the `FairCOCCO Score` that can be used to detect (un-)fairness, we move on to how it can be employed in order to obtain fair predictors. We focus on a standard supervised learning setup with the task to learn the map $\mathcal{X} \times \mathcal{A} \mapsto \mathcal{Y}$, with a given fairness condition $F$ that should be satisfied. Given a batch $\mathcal{D}$ of $N$ training triplets $\{(X_i, Y_i, A_i)\}^N_{i=1}$, a learning function $f_\theta(\cdot)$ with learnable parameters $\theta \in \Theta$, training loss $\mathcal{L}$, and $I_F$ denoting one of the fairness statistics from (9) that takes both the batch and learning function and returns the corresponding score, we arrive at a constrained optimization problem:

$$\min_{\theta \in \Theta} \frac{1}{N} \sum_{i=1}^{N} \mathcal{L}(f_\theta(x_i), y_i) \ \text{ subject to } \ I_F(\mathcal{D}, f_\theta) = 0 \qquad (10)$$

Practically speaking, this can be relaxed via a Lagrangian in order to obtain an unconstrained optimization problem that can be solved significantly more easily:

$$\min_{\theta \in \Theta} \frac{1}{N} \sum_{i=1}^{N} \mathcal{L}(f_\theta(x_i), y_i) + \lambda I_F(\mathcal{D}, f_\theta) \qquad (11)$$

The summary statistic (6) and therefore $I_F(\mathcal{D}, f_\theta)$ is differentiable and so as shown here can be employed as a regulariser in any gradient-based method, with $\lambda > 0$ a hyperparameter that determines the fairness-performance trade-off: a higher $\lambda$ guarantees higher fairness, but this typically leads to lower predictive performance. Consequently, this measure can be used to quantify and enforce fairness notions by controlling dependence between $\hat{Y}$, $A$ and $A$. We term this regularization scheme *FairCOCCO Learning*.

### 3.4 IN SUMMARY

The proposed kernel fairness measure provides a non-parametric, and strong characterization of fairness. The mappings allow both multivariate continuous and discrete variables to be embedded into the RKHS, from which we infer higher-order dependencies, and thus fairness effects. This enables the evaluation of multivariate, multitype fairness problems as commonly encountered in the real world. Additionally, the proposed metric and regularization methods are compatible with all dependency-based notions of fairness (as in Table 1), giving practitioners more flexibility in choosing the appropriate definitions for their scenario.

In our experiments, we use a Gaussian kernel: $k(X_i, X_j) = \exp\left(-\frac{||X_i - X_j||^2}{2\sigma^2}\right) \forall i, j \in N$ where $\sigma$, the bandwidth parameter, is selected with the median heuristic, $\sigma = \text{median}\{|x_i - x_j|, \forall i \neq j \in N\}$ (Schölkopf et al., 2002). As the calculation of (9) comprises a matrix inversion operation, the computational complexity scales with the number of samples $\mathcal{O}(N^3)$. We improve the scaling with training samples in two ways, *(1)* by employing a low-rank Cholesky decomposition of the Gram matrix (of rank $r$), resulting in $\mathcal{O}(r^2 N)$ complexity (Harbrecht et al., 2012) and *(2)* by estimating regulariser on mini-batches. We empirically investigate the effect of these relaxations on fairness estimation in Appendix C.2 and demonstrate that they lead to strong results in real-world experiments.

## 4 EXPERIMENTAL DEMONSTRATION

We now turn our attention to how our proposed methods works in practice. We perform experiments within the EO framework, since it is usually considered the most challenging, and it covers the middle-ground between the strict DP and lenient FTU definitions. However, we re-iterate that our method is *framework-agnostic* and attach further results under alternative definitions in Appendix C.1. There are a number of areas that require empirical demonstration, and so we proceed as follows:

1. First, in Section 4.1, we employ standard real-world benchmarks to compare against existing methods on **single binary attributes** and **outcomes**, resulting in competitive (and usually superior) predictive performance on these tasks while consistently producing the best DEO score.
2. Then, in Section 4.2, we apply `FairCOCCO` to real data with **multiple attributes** and **continuous outcomes**. This is an area that to the best of our knowledge no other method naturally extends to, and one that `FairCOCCO` now sets a strong benchmark for future work.
3. Finally, in Section 4.3, we consider the more complicated setting of fair learning in **image data** and **time series**. Here, we demonstrate how the problems of sepsis treatment and facial recognition are important applications of our method.

In the interest of limited space, we attach additional results in Appendix C. Specifically, we include experiments on: 4. **Different notions of fairness**: evaluating accuracy-fairness trade-off on different definitions of fairness (specifically DP and CAL); 5. **Statistical testing**: demonstrating the `FairCOCCO Score` as a test statistic for stronger fairness transparency; 6. **Sensitivity analysis**: to better evaluate the performance of our method on varying numbers of sensitive attributes.

**Benchmarks.** We compare against state-of-the-art fairness methods, including classic baselines (Zafar et al., 2017; Hardt et al., 2016; Donini et al., 2018) and more recent methods that adopt a stronger fairness quantification: `FACL` (Mary et al., 2019) and `FARMI` (Steinberg et al., 2020b), which leverages MCC and MI, respectively.

**Datasets.** Following the experiment design in recent works (Hardt et al., 2016; Donini et al., 2018), we employ 9 real-world datasets from the UCI machine learning repository (Dua & Graff, 2017). Specifically, we consider 4 datasets contain single sensitive attributes and binary outcomes and 5 datasets with multiple sensitive attributes and outcome of arbitrary type. We employ datasets with

Table 3: **Performance in binary setting.** Accuracy (ACC) and DEO on benchmark datasets. *NN* is an unregularised neural network, on top of which the regularizers from competitor methods and `FairCOCCO` are applied to. Best results are emboldened.

| Method | COMPAS | | German | | Drug | | Adult | |
|---|---|---|---|---|---|---|---|---|
| | ACC | DEO | ACC | DEO | ACC | DEO | ACC | DEO |
| Zafar et al. (2017) | $0.69 \pm 0.02$ | $0.10 \pm 0.06$ | $0.62 \pm 0.09$ | $0.13 \pm 0.11$ | $0.69 \pm 0.03$ | $0.02 \pm 0.07$ | 0.78 | 0.05 |
| Hardt et al. (2016) | $0.71 \pm 0.01$ | $0.08 \pm 0.01$ | $0.71 \pm 0.03$ | $0.11 \pm 0.18$ | $0.75 \pm 0.11$ | $0.14 \pm 0.08$ | 0.82 | 0.11 |
| Donini et al. (2018) | $0.73 \pm 0.01$ | $0.05 \pm 0.03$ | $0.73 \pm 0.04$ | $0.05 \pm 0.03$ | $0.80 \pm 0.03$ | $0.07 \pm 0.05$ | 0.81 | 0.01 |
| *NN* | $0.90 \pm 0.02$ | $0.06 \pm 0.00$ | $0.74 \pm 0.07$ | $0.11 \pm 0.35$ | $0.80 \pm 0.08$ | $0.06 \pm 0.12$ | 0.84 | 0.19 |
| Mary et al. (2019) | $0.88 \pm 0.02$ | $0.04 \pm 0.01$ | $0.73 \pm 0.03$ | $0.07 \pm 0.15$ | $\mathbf{0.80 \pm 0.04}$ | $\mathbf{0.01 \pm 0.01}$ | 0.82 | 0.08 |
| Steinberg et al. (2020b) | $0.88 \pm 0.01$ | $0.03 \pm 0.01$ | $0.71 \pm 0.10$ | $0.09 \pm 0.14$ | $0.79 \pm 0.05$ | $0.04 \pm 0.02$ | 0.80 | 0.10 |
| `FairCOCCO` | $\mathbf{0.89 \pm 0.01}$ | $\mathbf{0.00 \pm 0.01}$ | $\mathbf{0.74 \pm 0.03}$ | $\mathbf{0.02 \pm 0.09}$ | $0.80 \pm 0.06$ | $0.02 \pm 0.01$ | **0.83** | **0.04** |

different number of samples (ranging from 649 to 299285) and different feature counts (ranging from 10 to 128) to gain a better understanding of our method's performance profile.

Additionally, we also employ time-series dataset on sepsis treatment from the MIMIC-III ICU database (Johnson et al., 2016a) and an image dataset CelebA (Liu et al., 2015) for face attribute recognition. We provide additional information about benchmarks, datasets, model design, hyper-parameters, and evaluation methods in Appendix B. For all results, we report mean $\pm$ std over 10 runs.

### 4.1 BINARY ATTRIBUTES AND OUTCOMES

While the focus of this work is on introducing practical methods for fairness in multitype, multivariate settings, we want to first prove that `FairCOCCO` is also competitive with state-of-the-art methods on problems with binary sensitive attributes and outcomes. We reproduce benchmarks based on UCI's Drugs, German, Adult and COMPAS datasets. We compare against representative methods in literature as well as a standard (unfair) neural network (NN). For strong fairness methods, specifically including our method, `FACL` and `FARMI`, we employ the same NN as the underlying predictive model to ensure comparability.[3] We report our results in Table 3. `FairCOCCO` achieves higher levels of fairness (lower DEO) while maintaining strong predictive accuracy on all datasets except Drug. We note that (Mary et al., 2019) is specifically tailored for settings with binary sensitive attribute and outcome, but our method is more generally applicable to settings with multitype, multivariate sensitive attributes.

### 4.2 CONTINUOUS ATTRIBUTES AND OUTCOMES

Next, we illustrate the main contributions of our work, by demonstrating `FairCOCCO` can protect fairness in settings involving multiple sensitive attributes and outcomes of arbitrary type. We employ Crimes and Communities (C&C), Credit Card, KDD-Census, Law School, and Students datasets from the UCI repository. We start by looking at protection of single continuous attributes, before examining the joint protection of multiple sensitive attributes.

**Single continuous attribute.** We compare our method against our closest competitors `FACL` and `FARMI`.

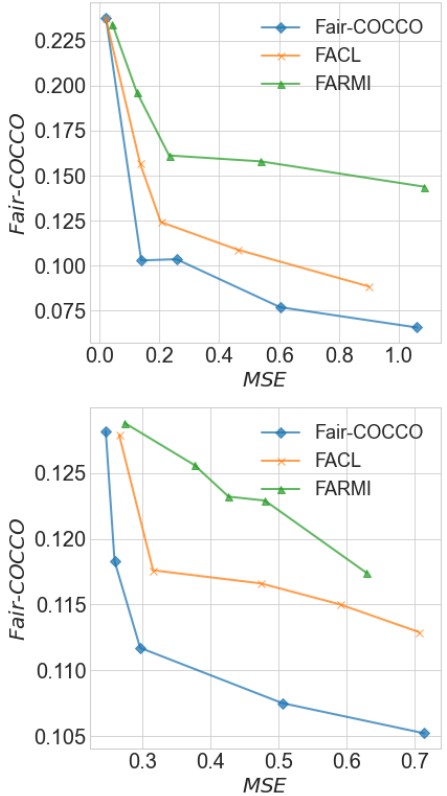

Figure 1: **Fairness accuracy trade-off. Crimes and Communities (top)** and **Students (bottom)**. Optimum desiderata at the origin, where both MSE and unfairness are minimized.

---

[3] We re-ran the available implementation in our own pipeline, reporting the best results between our re-runs and original reported scores.

Table 4: **Protection of multiple attributes.** Investigation on joint fairness effects and fairness protection with respect to individual sensitive attributes on array of benchmarks. Lowest MSE/ACC, `FairCOCCO` and DEO scores are emboldened.

### Crimes and Communities

| Method | Joint MSE | Joint COCCO | racePctBlack COCCO | racePctBlack DEO | racePctAsian COCCO | racePctAsian DEO | racePctHisp COCCO | racePctHisp DEO |
|---|---|---|---|---|---|---|---|---|
| *NN* | $0.22 \pm 0.01$ | $0.27 \pm 0.01$ | $0.24 \pm 0.02$ | $0.25 \pm 0.03$ | $0.10 \pm 0.01$ | $0.14 \pm 0.02$ | $0.16 \pm 0.01$ | $0.08 \pm 0.04$ |
| FACL | $0.66 \pm 0.01$ | $0.15 \pm 0.02$ | $0.10 \pm 0.02$ | $0.10 \pm 0.08$ | $0.10 \pm 0.02$ | $0.13 \pm 0.06$ | $0.09 \pm 0.02$ | $0.09 \pm 0.03$ |
| FARMI | $0.65 \pm 0.01$ | $0.20 \pm 0.02$ | $0.14 \pm 0.02$ | $0.10 \pm 0.06$ | $0.11 \pm 0.02$ | $\mathbf{0.11 \pm 0.05}$ | $0.13 \pm 0.02$ | $0.08 \pm 0.02$ |
| FairCOCCO | $\mathbf{0.63 \pm 0.01}$ | $\mathbf{0.11 \pm 0.01}$ | $\mathbf{0.08 \pm 0.01}$ | $\mathbf{0.07 \pm 0.05}$ | $\mathbf{0.07 \pm 0.01}$ | $0.11 \pm 0.05$ | $\mathbf{0.07 \pm 0.02}$ | $\mathbf{0.05 \pm 0.03}$ |

### KDD-Census

| Method | Joint ACC | Joint COCCO | age COCCO | age DEO | sex COCCO | sex DEO | race COCCO | race DEO |
|---|---|---|---|---|---|---|---|---|
| *NN* | $0.95 \pm 0.02$ | $0.18 \pm 0.04$ | $0.17 \pm 0.03$ | $0.24 \pm 0.06$ | $0.07 \pm 0.00$ | $0.09 \pm 0.01$ | $0.07 \pm 0.01$ | $0.10 \pm 0.02$ |
| FACL | $0.93 \pm 0.01$ | $0.10 \pm 0.02$ | $0.10 \pm 0.03$ | $0.12 \pm 0.03$ | $0.04 \pm 0.01$ | $0.03 \pm 0.00$ | $0.08 \pm 0.02$ | $0.09 \pm 0.02$ |
| FARMI | $0.88 \pm 0.03$ | $0.15 \pm 0.05$ | $0.13 \pm 0.05$ | $0.18 \pm 0.05$ | $0.05 \pm 0.00$ | $0.04 \pm 0.01$ | $0.07 \pm 0.01$ | $0.05 \pm 0.02$ |
| FairCOCCO | $\mathbf{0.94 \pm 0.02}$ | $\mathbf{0.02 \pm 0.00}$ | $\mathbf{0.02 \pm 0.00}$ | $\mathbf{0.01 \pm 0.00}$ | $\mathbf{0.00 \pm 0.01}$ | $\mathbf{0.02 \pm 0.01}$ | $\mathbf{0.00 \pm 0.01}$ | $\mathbf{0.02 \pm 0.00}$ |

### Credit Card

| Method | Joint ACC | Joint COCCO | sex COCCO | sex DEO | education COCCO | education DEO | marriage COCCO | marriage DEO |
|---|---|---|---|---|---|---|---|---|
| *NN* | $0.82 \pm 0.02$ | $0.13 \pm 0.01$ | $0.06 \pm 0.01$ | $0.08 \pm 0.00$ | $0.04 \pm 0.01$ | $0.02 \pm 0.00$ | $0.04 \pm 0.02$ | $0.03 \pm 0.02$ |
| FACL | $0.80 \pm 0.01$ | $0.07 \pm 0.00$ | $0.04 \pm 0.00$ | $0.03 \pm 0.00$ | $0.02 \pm 0.01$ | $0.02 \pm 0.00$ | $0.05 \pm 0.01$ | $0.03 \pm 0.01$ |
| FARMI | $0.81 \pm 0.02$ | $0.05 \pm 0.00$ | $0.02 \pm 0.00$ | $0.02 \pm 0.00$ | $0.03 \pm 0.00$ | $0.02 \pm 0.00$ | $0.04 \pm 0.01$ | $0.03 \pm 0.01$ |
| FairCOCCO | $\mathbf{0.81 \pm 0.01}$ | $\mathbf{0.01 \pm 0.00}$ | $\mathbf{0.00 \pm 0.00}$ | $\mathbf{0.01 \pm 0.00}$ | $\mathbf{0.00 \pm 0.00}$ | $0.02 \pm 0.00$ | $\mathbf{0.00 \pm 0.00}$ | $\mathbf{0.01 \pm 0.00}$ |

### Law School

| Method | Joint ACC | Joint COCCO | male COCCO | male DEO | race COCCO | race DEO |
|---|---|---|---|---|---|---|
| *NN* | $0.89 \pm 0.03$ | $0.07 \pm 0.04$ | $0.01 \pm 0.00$ | $0.02 \pm 0.00$ | $0.11 \pm 0.01$ | $0.12 \pm 0.05$ |
| FACL | $0.85 \pm 0.02$ | $0.04 \pm 0.02$ | $0.00 \pm 0.01$ | $0.01 \pm 0.00$ | $0.04 \pm 0.02$ | $0.05 \pm 0.03$ |
| FARMI | $0.86 \pm 0.02$ | $0.04 \pm 0.01$ | $0.01 \pm 0.00$ | $0.01 \pm 0.00$ | $0.03 \pm 0.01$ | $0.04 \pm 0.02$ |
| FairCOCCO | $\mathbf{0.89 \pm 0.01}$ | $\mathbf{0.02 \pm 0.00}$ | $\mathbf{0.00 \pm 0.00}$ | $\mathbf{0.00 \pm 0.00}$ | $\mathbf{0.01 \pm 0.01}$ | $\mathbf{0.04 \pm 0.00}$ |

### Students

| Method | Joint MSE | Joint COCCO | age COCCO | age DEO | sex COCCO | sex DEO |
|---|---|---|---|---|---|---|
| *NN* | $0.25 \pm 0.05$ | $0.16 \pm 0.03$ | $0.12 \pm 0.02$ | $0.06 \pm 0.03$ | $0.09 \pm 0.03$ | $0.07 \pm 0.02$ |
| FACL | $0.32 \pm 0.03$ | $0.14 \pm 0.02$ | $0.12 \pm 0.02$ | $0.07 \pm 0.03$ | $0.07 \pm 0.02$ | $0.04 \pm 0.05$ |
| FARMI | $0.36 \pm 0.06$ | $0.15 \pm 0.01$ | $0.11 \pm 0.02$ | $\mathbf{0.05 \pm 0.01}$ | $0.10 \pm 0.02$ | $0.07 \pm 0.04$ |
| FairCOCCO | $\mathbf{0.29 \pm 0.05}$ | $\mathbf{0.14 \pm 0.02}$ | $\mathbf{0.10 \pm 0.03}$ | $0.06 \pm 0.03$ | $\mathbf{0.07 \pm 0.01}$ | $\mathbf{0.03 \pm 0.03}$ |

While FACL does not support multiple attributes, it is applicable to protect a single continuous variable. FARMI is only compatible with discrete sensitive attributes; we thus binarise the sensitive attributes at the median during training. We take the datasets C&C and Students, and use protected attributes racePctBlack and age respectively. We plot the performance versus fairness by varying the fairness penalty in Figure 1. Notably, FairCOCCO obtains a better trade-off between fairness and MSE than both methods (optimum desiderata at the origin).

**Multiple (arbitrary type) attributes.** Going one step further, we want to evaluate the concurrent protection of multiple sensitive attributes. We note that while this is natural for FairCOCCO, to the best of our knowledge, there are no existing methods that can jointly protect multiple sensitive attributes of arbitrary type. To enable adequate comparison, we adapt FACL and FARMI by including a separate regularization term for each attribute. In contrast, the FairCOCCO regularization is applied directly and jointly on all sensitive attributes. Previously, we showed that the protection of individual fairness effects does not guarantee protection of joint fairness. To that end, we are interested in analyzing both joint fairness effects and protection w.r.t. individual attributes. In Table 4, we evaluate the joint fairness (**Joint**) and fairness on individual attributes (e.g. **racePctBlack**, **racePctAsian**, **racePctHisp** on **C&C**). To evaluate individual fairness, we also calculate the DEO by binarising the attributes at the median during evaluation.

We first note that FairCOCCO and DEO scores are highly correlated in their respective estimation of unfairness. However, the key result we wish to highlight is that not only does FairCOCCO successfully minimize joint fairness effects, it also consistently minimizes the levels of unfairness for each sensitive attribute. The same cannot be said for FARMI and FACL, where the joint fairness outcomes are inadequate as the protection granted to individual attributes are traded-off to the detriment of other attributes. To better investigate the sensitivity of our method to the number of sensitive attributes, the performance fairness trade-off by varying the number of protected attributes in Appendix C.4.

Table 5: **Facial attribute recognition**. Accuracy (ACC) and DEO on three separate classification tasks - **attractive**, **smile**, and **wavy hair**. Best results are emboldened.

| Method | attractive | | smile | | wavy hair | |
|---|---|---|---|---|---|---|
| | ACC | DEO | ACC | DEO | ACC | DEO |
| *NN* | $0.82 \pm 0.02$ | $0.43 \pm 0.03$ | $0.98 \pm 0.03$ | $0.05 \pm 0.01$ | $0.81 \pm 0.02$ | $0.18 \pm 0.02$ |
| FACL | $0.78 \pm 0.02$ | $0.11 \pm 0.02$ | $0.95 \pm 0.03$ | $0.02 \pm 0.00$ | $0.78 \pm 0.02$ | $0.10 \pm 0.01$ |
| FARMI | $0.79 \pm 0.03$ | $0.07 \pm 0.01$ | $\mathbf{0.96 \pm 0.02}$ | $\mathbf{0.01 \pm 0.00}$ | $0.74 \pm 0.01$ | $0.04 \pm 0.00$ |
| FairCOCCO | $\mathbf{0.80 \pm 0.03}$ | $\mathbf{0.03 \pm 0.00}$ | $0.96 \pm 0.04$ | $\mathbf{0.01 \pm 0.00}$ | $\mathbf{0.80 \pm 0.02}$ | $\mathbf{0.02 \pm 0.00}$ |

### 4.3 BEYOND TABULAR DATA

**CelebA facial attributes recognition.** In this section, we highlight that `FairCOCCO` can be applied beyond tabular data by experimenting on the CelebA dataset (Liu et al., 2015). The CelebA dataset contains images of celebrity faces, where each face is associated with binary sensitive attributes, including gender. We follow the experimental design in (Chuang & Mroueh, 2021) and form binary classification tasks using attributes *attractive*, *smile*, and *wavy hair*, and treat gender as the sensitive attribute. We fine-tune a ResNet-18 (He et al., 2016) with two additional hidden layers to perform the classification task. We report the results in Table 5, noting similar improvements in fairness with little decrease in accuracy, especially on the classifying *attractive* and *wavy hair*.

**Sepsis treatment.** Finally, we emphasize that `FairCOCCO` is not limited to the standard supervised learning setup and demonstrate how our approach can be applied for learning fairer policies in time series setting. We employ the MIMIC-III ICU database (Johnson et al., 2016a), containing data routinely collected from adult patients in the United States. We analyze the decisions made by clinicians to treat sepsis, using a patient cohort fulfilling the Sepsis-3 criteria,

Table 6: **Sepsis treamtent.** Accuracy (ACC), DEO and `FairCOCCO` score on learning fair sepsis treatment policies; the best results are emboldened.

| Method | ACC | DEO | COCCO |
|---|---|---|---|
| *NN* | $0.82 \pm 0.03$ | $0.05 \pm 0.03$ | $0.13 \pm 0.02$ |
| FACL | $0.81 \pm 0.04$ | $0.02 \pm 0.01$ | $0.08 \pm 0.02$ |
| FARMI | $0.78 \pm 0.04$ | $0.03 \pm 0.01$ | $0.10 \pm 0.01$ |
| FairCOCCO | $\mathbf{0.81 \pm 0.02}$ | $\mathbf{0.00 \pm 0.01}$ | $\mathbf{0.04 \pm 0.01}$ |

delineated by Komorowski et al. (2018). For each patient, we have relevant physiological parameters recorded at 4 hour resolution, and static demographic context. The task is to predict the clinical intervention to treat sepsis by learning from clinician's actions. For this, we have access to a binary variable corresponding to clinical interventions targeting sepsis. Ground-truth treatment outcomes are computed from SOFA scores (measuring sequential organ failure) and lactate levels (correlated with severity of sepsis) in the subsequent time step, and we consider gender as the sensitive attribute. For the complete problem setup, refer to the Appendix B.2. Table 6 indicates that `FairCOCCO` successfully reduced any bias contained in expert demonstrations and achieved the best predictive and fair performance when compared to `FACL` and `FARMI`.

## 5 DISCUSSION

In this work, we proposed `FairCOCCO`, a kernel-based fairness measure that strongly quantifies the level of unfairness in the presence of multiple sensitive attributes of mixed type. Specifically, we introduced a normalized fairness metric (`FairCOCCO Score`), applicable to different problem settings and dependency-based fairness notions, and a fairness regularization scheme. Through our experiments, we empirically demonstrated superior fairness-prediction trade-off and protection of multiple and individual fairness outcomes.

**Limitations and future works.** The main limitation of our work is computational complexity—the matrix operations, required to kernalise the data and embed it in the RKHS, has complexity $\mathcal{O}(N^3)$. We propose two directions to alleviate this (i.e. low-rank approximation, mini-batch evaluations), which empirically do not noticeably impact performance. Future works should consider speeding up kernel operations using methods proposed in (Zhang et al., 2012). Additionally, while our regularizer can be applied to models trained using gradient-based methods, future works should extend our approach to be compatible with powerful decision-tree based algorithms.

## ETHICS AND REPRODUCIBILITY STATEMENT

**Ethics statement.** We caution against using our proposed methods as a *certificate* of fairness. As Corbett-Davies et al. (2017) rightfully emphasize, fairness measures do not rule out unfair practices. Additionally, future works should focus on interpretable fairness quantification that sheds insight on root causes of unfairness, allowing them to be eliminated through procedural changes rather than solely in prediction tasks. Lastly, we encourage more lively discourse on philosophical implications of ML methods on justice and fairness (Kuppler et al., 2021) that is critical to Fair ML deployment.

**Reproducibility statement.** We detailed exact implementation details, including dataset preprocessing, implementation of benchmark methods, architecture design, hyperparameter tuning, and evaluation methods in Section 3, Section 4 and Appendix B. We will release code upon acceptance of the paper for the camera-ready version.

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

# A  MORE ON FAIRCOCCO

## A.1  CLOSED FORM EXPRESSION

We introduced covariance operators on RKHSs, which can be used to quantify unconditional $\left(V_{\hat{Y}A}\right)$ and conditional fairness $\left(V_{\hat{Y}\ddot{A}\,|\,Y}\right)$. FairCOCCO is based on the Hilbert-Schmidt (HS) norm of the covariance operators. An operator $A : \mathcal{H}_1 \rightarrow \mathcal{H}_2$ is called HS if, for complete orthonormal systems $\{\phi_i\}$ of $\mathcal{H}_1$ and $\{\psi_j\}$ of $\mathcal{H}_2$, the sum $\sum_{i,j}\langle \psi_j, A\phi_i\rangle_{HS}^2$ is finite (Reed & Simon, 1980). Thus, for an HS operator $A$, the HS norm, $||A||_{HS}$ is defined as $||A||_{HS}^2 = \sum_{i,j}\langle \psi_j, A\phi_i\rangle_{HS}^2$. Provided that $V_{\hat{Y}\ddot{A}\,|\,Y}$ and $V_{\hat{Y}A}$ are HS operators, FairCOCCO scores can be expressed as:

$$||V_{\hat{Y}\ddot{A}\,|\,Y}||_{HS}^2 \quad \text{(conditional fairness measure)}$$

$$||V_{\hat{Y}A}||_{HS}^2 \quad \text{(unconditional fairness measure)}$$

The umlaut on $A$ represent extended variable sets, i.e. $\ddot{A} = (A, Y)$. Here, we briefly flesh out the closed-form expression of the empirical estimators, while more details can be found at (Fukumizu et al., 2007; Gretton et al., 2005). Let $G_Y$ be the centered Gram matrices, such that:

$$G_{Y,ij} = \left\langle k_{\mathcal{Y}}(\cdot, Y_i) - \hat{m}_Y^{(N)}, k_{\mathcal{Y}}(\cdot, Y_j) - \hat{m}_Y^{(N)} \right\rangle_{\mathcal{H}_{\mathcal{Y}}}$$

We choose a Gaussian RBF kernel, $k(Y_i, Y_j) = \exp\left(-\frac{||Y_i - Y_j||^2}{2\sigma^2}\right) \ \forall\, i, j \in N$, and employ the median heuristic introduced by Schölkopf et al. (2002), i.e. $\sigma = median\{|Y_i - Y_j|, \forall\, i \neq j \in N\}$ to select bandwidth $\sigma$. Additionally, $\hat{m}_Y^{(N)} = 1/N \sum_{i=1}^N k_{\mathcal{Y}}(\cdot, Y_i)$ is the empirical mean. $G_A, G_{\hat{Y}}$ are defined similarly. Based on this, proxy Gram matrices $R_Y$ can be defined as follows:

$$R_Y = G_Y(G_Y + \epsilon N I_N)^{-1}$$

where $\epsilon = 1\mathrm{e}{-4}$ is a regularization constant, used in the same way as Bach & Jordan (2002), $I_N$ is an identity matrix and $R_{\hat{Y}}, R_A$ are defined similarly. The empirical estimator of $||\hat{V}_{\hat{Y}\ddot{A}\,|\,Y}||_{HS}^2$ can then be computed:

$$\hat{I} = ||\hat{V}_{\hat{Y}\ddot{A}\,|\,Y}||_{HS}^2 \tag{12}$$

$$= \mathrm{Tr}[R_{\hat{Y}}R_{\ddot{A}} - 2R_{\hat{Y}}R_{\ddot{A}}R_Y + R_{\hat{Y}}R_Y R_{\ddot{A}}R_Y] \tag{13}$$

The unconditional fairness score can similarly be estimated empirically as follows (note that unconditional dependence does not entail using extended variables):

$$\hat{I} = ||\hat{V}_{\hat{Y}A}||_{HS}^2 \tag{14}$$

$$= \mathrm{Tr}[R_{\hat{Y}}R_A] \tag{15}$$

**Choice of Kernels.** While, in general, kernel dependence measures depend not only on variable distributions, but also the choice of kernel, Fukumizu et al. (2007) showed that, in the limit of infinite data and assumptions on richness of the RKHS, the estimates converges to a kernel-independent value. We employ a Gaussian RBF (characteristic kernel) in our experiments.

**On the computational complexity.** For our experiments, we use a Gaussian RBF kernel: $k(X_i, X_j) = \exp\left(-\frac{||X_i - X_j||^2}{2\sigma^2}\right) \ \forall\, i, j \in N$ where $\sigma$ is the tuneable bandwidth parameter. We employ the median heuristic introduced by Schölkopf et al. (2002), i.e. $\sigma = median\{|x_i - x_j|, \forall\, i \neq j \in N\}$ to select bandwidth.

As the calculation of (9) comprises a matrix inversion operation, the computational complexity scales with the number of samples in $\mathcal{O}(N^3)$. We improve the scaling with training samples in two ways, *(1)* by employing a low-rank Cholesky decomposition of the Gram matrix (of rank $r$), resulting in $\mathcal{O}(r^2 N)$ complexity (Harbrecht et al., 2012) and *(2)* by estimating regulariser on mini-batches. We empirically demonstrate that these lead to strong results in real-world experiments.

## A.2 FairCOCCO Score

Here, we derive `FairCOCCO` score from the underlying measure using the Cauchy-Schwarz Inequality. The `FairCOCCO` score for conditional fairness and unconditional fairness can be written as:

$$\texttt{FairCOCCO Score (unconditional)} = \frac{||V_{\hat{Y}A}||_{HS}^2}{||V_{\hat{Y}\hat{Y}}||_{HS}||V_{AA}||_{HS}}$$

$$\texttt{FairCOCCO Score (conditional)} = \frac{||V_{\hat{Y}\breve{A}\,|\,Y}||_{HS}^2}{||R_{\hat{Y}} - R_{\hat{Y}}R_Y||_{HS}||R_{\breve{A}} - R_{\breve{A}}R_Y||_{HS}}$$

We start by looking unconditional version of `FairCOCCO`, we know from (14) and the Cauchy-Schwarz inequality for the inner-product $\langle \cdot, \cdot \rangle$ that:

$$|||\hat{V}_{\hat{Y}A}||_{HS}^2| = |\text{Tr}[R_{\hat{Y}}R_A]| = |\langle R_{\hat{Y}}^T, R_A \rangle|$$
$$\leq ||R_{\hat{Y}}||_{HS}||R_A||_{HS} = \sqrt{\text{Tr}[R_{\hat{Y}}^T R_{\hat{Y}}]}\sqrt{\text{Tr}[R_A^T R_A]}$$
$$= ||\hat{V}_{\hat{Y}\hat{Y}}||_{HS}||\hat{V}_{AA}||_{HS}$$

By the inequality, `FairCOCCO` Score (unconditional) $\in [-1, 1]$. Additionally, as the score is also non-negative, it takes value $\in [0, 1]$ where 0 indicates perfect fairness (as indicated by Lemma 3.1). By contrast, the score takes value 1 iff the gram matrices, $R_{\hat{Y}}$ and $R_A$, are linearly dependent (i.e. perfectly unfair). The derivation and interpretation can similarly be shown for the conditional case:

$$|||\hat{V}_{\hat{Y}A\,|\,Y}||_{HS}^2| = |\text{Tr}[R_{\hat{Y}}R_A - 2R_{\hat{Y}}R_A R_Y + R_{\hat{Y}}R_Y R_A R_Y]|$$
$$= |\text{Tr}[(R_{\hat{Y}} - R_{\hat{Y}}R_Y)(R_A - R_A R_Y)]| = |\langle (R_{\hat{Y}} - R_{\hat{Y}}R_Y)^T, (R_A - R_A R_Y) \rangle|$$
$$\leq ||R_{\hat{Y}} - R_{\hat{Y}}R_Y||_{HS}||R_A - R_A R_Y||_{HS}$$

Here, $R_{\hat{Y}} - R_{\hat{Y}}R_Y$ is related to the conditional covariance operator, i.e. $\hat{V}_{\hat{Y}\hat{Y}\,|\,Y}$, which captures the conditional covariance of $\hat{Y}$ given $Y$. See (Fukumizu et al., 2007; 2009; Baker, 1973) and others for more.

# B  Experimental Details

## B.1  Supervised Learning Tasks

### B.1.1  Model Details

For all experiments, we train a two-layer neural network with ReLU-activated nodes. The number of nodes chosen is between $40{\sim}100$ depending on the complexity of the data. The network is trained with Cross Entropy or MSE Loss and is optimized using Adam (Kingma & Ba, 2014). The hyperparameters include batch size $\in \{64, 128, 256\}$, learning rate $\in \{1e{-}2, 1e{-}3, 1e{-}4\}$, and fairness penalty $\in \{0.0, 0.5, 1.0, 2.0, 5.0\}$ and are chosen through cross-validation. For datasets without a defined test set, the data is split 60-20-20 into train, validation and test set and results are averaged over 10 runs. Experiments are run on either a CPU or NVIDIA Tesla K40C GPU, taking around an hour.

### B.1.2  Datasets

**Adult** (Kohavi, 1996). The task on the Adult dataset is to classify whether an individual's income exceeded \$50K/year based on census data. There are 48842 training instances and 14 attributes, 4 of which are sensitive attributes (`age`, `race`, `sex`, `native-country`). Here, the sensitive attribute is chosen to be `sex`, which can be either female or male.

**Drug Consumption (Drugs)** (Mirkes, 2015). The classification problem is whether an individual consumed drugs based on personality traits. The dataset contains 1885 respondents and 12 personality measurements. Respondents are questioned on drug use on 18 drugs, including a fictitious drug `Semeron` to identify over-claimers. Here, we focus on `Heroin` use, drop the respondents who

Table 7: **Description of datasets.** '-B' suffix indicates binary variables, '-D' indicates discrete variables (i.e. >2 classes) '-C' indicates continuous variables.

|  | Dataset | Examples | Features | Sensitive ($A$) | Outcome ($Y$) |
|---|---|---|---|---|---|
| **Single sensitive attributes** | **Adult** | 45222 | 12 | Gender-B | Income-B |
|  | **Drugs** | 1885 | 11 | Ethnicity-B | Drug use-B |
|  | **German** | 1700 | 20 | Foreign-B | Income-B |
|  | **COMPAS** | 6172 | 10 | Ethnicity -B | Recidivism-B |
| **Multiple sensitive attributes** | **C&C** | 1994 | 128 | Ethnicity-C ($\times 4$) | Crime rate-C |
|  | **Students** | 649 | 33 | Age-C, Gender-B | Performance-C |
|  | **KDD-Census** | 299285 | 40 | Sex-B, Race-B, Age-C | Income-B |
|  | **Credit Card** | 30000 | 24 | Sex-B, Marriage-D, Education-D | Default-B |
|  | **Law School** | 20798 | 12 | Male-B, Race-D | Pass-B |

claimed to use `Semeron` and transform the categorical response into a binary outcome: "Never Used" versus "Used". The binary sensitive attribute is `Ethnicity`.

**South German Credit (German)** (Hoffman, 1994). The German dataset contains 1000 instances with 20 predictor variables of a debtor's financial history and demographic information, which are used to predict binary credit risk (i.e. complied with credit contract or not). The sensitive attribute is a binary variable indicating whether the debtor is of foreign nationality.

**COMPAS** (Angwin et al., 2016). COMPAS is a commercial software commonly used by judges and parole officers for scoring a criminal defendant's likelihood of recidivism. The dataset contains 6172 instances with 10 features. The outcome is a binary variable corresponding to whether violent recidivism occurred (`is_violent_recid`) and the sensitive attribute is `race`, which is binarised into "Caucasian" and "Non-Caucasian" defendants.

**Communities and Crime (C&C)** (Redmond, 2009). C&C contains socio-economic data from the 1990 US Census, law enforcement data from the 1990 US LEMAS survey and crime data from 1995 FBI UCR. It contains 1994 instances of communities with 128 attributes. The outcome of the regression problem is crime rate within each community `ViolentCrimesPerPop`, which is a continuous value. There are three sensitive attributes, corresponding to ethnic proportions in the community—`racePctBlack`, `racePctWhite`, `racePctAsian`.

**Student Performance (Students)** (Cortez, 2014). The Students dataset predicts academic performance in the last year of high school. There are 649 instances with 33 attributes, including past academic information and student demographics. The response variable is a continuous variable corresponding to final grade and the sensitive attributes are `age` (continuous value from 15-22) and `sex` ('F'-female, 'M'-male).

### B.2    TIME SERIES TASK

The data used to develop and evaluate our experiment on fair imitation learning is extracted from the MIMIC-III ICU database (Johnson et al., 2016a), based on the Sepsis-3 cohort defined by Komorowski et al. (2018).

**Discrimination in Healthcare.** Sepsis is one of the leading causes of mortality in intensive care units (Singer et al., 2016), and while efforts have been made to provide clinical guidelines for treatment, physicians at the bedside largely rely on experience, giving rise to possible variations in fair treatments. Prejudice in healthcare has been reported in many instances—for example, healthcare professionals are more likely to downplay women's health concerns (Rogers & Ballantyne, 2008) and racial biases affect pain assessment and treatment prescribed (Hoffman et al., 2016). Thus, it is critical, when learning to imitate expert policy, that no underlying prejudices are leaked into the learned policy.

**Problem Setup.** We have access to a set of expert trajectories $\mathcal{D} = \{\tau_1, ..., \tau_N\}$, where each trajectory is a sequence of state-action pairs $\{(s_1, a_1), ..., (s_T, a_T)\}$. The time-varying state space is modelled with a Markov Decision Process (MDP), i.e. at every time step $t$, the agent observes current state $s_t$ and takes action $a_t$.

**Data.** We obtain data from MIMIC-III and use the pre-processing scripts provided by Komorowski et al. (2018) to extract patients satisfying the Sepsis-3 criteria. For each patient, we have relevant physiological parameters, including demographics, lab values, vital signs and intake/output events. Data are aggregated into 4 hour windows.

**State Space.** The pre-processing yields $45 \times 1$ feature vectors for each patient at each time step, which are summarized in Table 8. We consider gender as the sensitive attribute.

Table 8: **MIMIC-III Features.** Description of patient features recorded at four hour intervals.

| Feature Type | Features |
|---|---|
| Demographic | Gender, Age, Weight (kg), |
| Static | Re-admission, Glasgow Coma Scale (GCS), Sequential Organ Failure Assessment (SOFA), Systematic Inflammatory Response Syndrome (SIRS), Shock Index, |
| Lab Values | Potassium, Sodium, Chloride, Glucose, Magnesium, Calcium, White Blood Cell Count, Platelets Count, Bicarbonate, Hemoglobin, Partial Thromboplastin Time (PTT), Prothrombin Time (PT), Arterial pH, Arterial Blood Gas, Arterial Lactate, Blood Urea Nitrogen (BUN), Creatinine, Serum Glutamic-Oxaloacetic Transaminase (SGOT), Serum Glutamic-Pyruvic Transaminase (SGPT), Total Bilirubin, International Normalized Ratio (INR), |
| Vitals | Heart Rate, Systolic Blood Pressure, Mean Blood Pressure, Diastolic Blood Pressure, Respiratory Rate, Temperature (Celsius), FiO2, PaO2, PaCO2, PaO2/FiO2 ratio, SpO2, |
| Intake/Output | Mechanical Ventilation, Fluid Intake (4 hourly), Fluid Intake (Total), Fluid Output (4 hourly), Fluid Output (Total) |

**Action Space.** We define a binary action for medical intervention based on intravenous (IV) fluid and maximum vasopressor (VP) dosage in a given 4 hour window, where $a_t = 1$ represent either or both interventions taken, and $a_t = 0$ indicates no action taken.

**Treatment Outcome.** The ground truth treatment outcome in each time step is evaluated using SOFA (measuring organ failure) and the arterial lactate levels (higher in septic patients). Specifically, the treatment outcome penalizes high SOFA scores and increases in SOFA and lactate levels from the previous time step (Raghu et al., 2017):

$$Y_t = -0.025\mathbb{1}(s_{t+1}^{SOFA} = s_t^{SOFA} \,\&\, s_{t+1}^{SOFA} > 0) - 0.125(s_{t+1}^{SOFA} - s_T^{SOFA})$$
$$- 2\tanh(s_{t+1}^{lactate} - s_t^{lactate})$$

**Behavioral Cloning.** Our proposed framework should work with any imitation learning algorithm as long as predictions of action rewards are differentiable. For now, we will focus on behavioral cloning. The expert's demonstrations $\mathcal{D}$ are divided into i.i.d. state-action pairs. We train a neural network as described in the experimental setup to predict posterior action probabilities.

## C  ADDITIONAL EXPERIMENTS

In this section, we provide additional results to comprehensively evaluate our proposed methods, specifically:

1. **DP and EO**: While the main paper investigates fairness using EO, Appendix C.1 demonstrates application of `FairCOCCO` using DP and CAL notions of fairness.
2. **Estimation convergence**: Appendix C.2 evaluates the convergence of `FairCOCCO score` estimation on different mini-batch sizes on real datasets.
3. **Statistical testing**: Appendix C.3 demonstrates how the `FairCOCCO Score` can be employed as a test statistic in permutation-based testing for stronger fairness transparency.
4. **Sensitivity**: Appendix C.4 investigates performance sensitivities, specifically performance-fairness trade-offs, according to varying numbers of sensitive attributes.

### C.1  ADDITIONAL RESULTS: EXPERIMENTS WITH DP AND CAL

To highlight `FairCOCCO`'s compatibility with fairness definitions other than EO, we apply it to demographic parity (DP) and calibration (CAL). We perform the same experiments on 1) binary classification tasks, 2) regression task with multiple sensitive attributes. The experiments are performed using the procedures described in the experimental setup.

**Demographic Parity.** DP requires statistical independence between predictions and attributes. *Disparate impact* (DI) is a metric frequently used to evaluate DP (Feldman et al., 2015):

$$DI = \frac{P(\hat{Y} = 1 | A = 1)}{P(\hat{Y} = 1 | A = 0)} \tag{16}$$

where $A = 1$ and $A = 0$ denote respectively the discriminated and non-discriminated groups. The US Equal Employment Opportunity Commission Recommendation advocates that DI should not be below $80\%$, commonly known as the $80\%$-rule.[4] DI closer to $1$ corresponds to lower levels of disparate impacts across population subgroups. We show the performance of `FairCOCCO` for DP in Table 9 and 10, demonstrating superior performance on a benchmark of binary classification tasks as well as protection of multiple sensitive attributes in regression settings.

Table 9: **Performance in binary setting.** Accuracy (ACC) and DI under DP. *NN* is an unregularised neural network that is used as base learner; the best results are emboldened.

| Method | COMPAS | | German | | Drug | | Adult | |
|---|---|---|---|---|---|---|---|---|
| | ACC | DI | ACC | DI | ACC | DI | ACC | DI |
| Donini et al. (2018) | $0.70 \pm 0.02$ | $0.81 \pm 0.03$ | $0.70 \pm 0.06$ | $0.93 \pm 0.07$ | $0.74 \pm 0.03$ | $0.75 \pm 0.01$ | 0.72 | 0.84 |
| NN | $0.90 \pm 0.02$ | $0.39 \pm 0.32$ | $0.74 \pm 0.07$ | $1.26 \pm 0.54$ | $0.80 \pm 0.08$ | $0.42 \pm 0.22$ | 0.84 | 0.22 |
| Mary et al. (2019) | $0.87 \pm 0.04$ | $0.76 \pm 0.07$ | $0.71 \pm 0.08$ | $0.96 \pm 0.25$ | $0.80 \pm 0.06$ | $0.73 \pm 0.17$ | 0.79 | 0.83 |
| Steinberg et al. (2020b) | $0.86 \pm 0.03$ | $0.83 \pm 0.05$ | $0.71 \pm 0.06$ | $0.93 \pm 0.13$ | $0.77 \pm 0.03$ | $0.86 \pm 0.05$ | 0.77 | 0.76 |
| `FairCOCCO` | $\mathbf{0.88 \pm 0.03}$ | $\mathbf{0.90 \pm 0.06}$ | $0.73 \pm 0.06$ | $\mathbf{1.02 \pm 0.19}$ | $\mathbf{0.78 \pm 0.02}$ | $\mathbf{0.84 \pm 0.07}$ | **0.83** | **0.97** |

Table 10: **Protection of multiple attributes.** Level of protection provided to individual attributes when all attributes are simultaneously protected under DP. Lowest MSE & `FairCOCCO` scores are emboldened. **(left)** C&C dataset, **(right)** Students dataset.

| Method | Joint | | racePctBlack | racePctWhite | racePctAsian | racePctHisp | Method | Joint | | age | sex |
|---|---|---|---|---|---|---|---|---|---|---|---|
| | MSE | COCCO | COCCO | COCCO | COCCO | COCCO | | MSE | COCCO | COCCO | COCCO |
| NN | 0.22 $\pm 0.01$ | 0.20 $\pm 0.08$ | 0.16 $\pm 0.06$ | 0.24 $\pm 0.03$ | 0.03 $\pm 0.01$ | 0.09 $\pm 0.05$ | NN | 0.25 $\pm 0.05$ | 0.16 $\pm 0.06$ | 0.13 $\pm 0.03$ | 0.11 $\pm 0.07$ |
| FACL | 0.53 $\pm 0.04$ | 0.09 $\pm 0.02$ | 0.07 $\pm 0.01$ | 0.15 $\pm 0.04$ | 0.05 $\pm 0.03$ | 0.07 $\pm 0.02$ | FACL | 0.30 $\pm 0.02$ | 0.08 $\pm 0.01$ | 0.04 $\pm 0.01$ | **0.03** $\pm \mathbf{0.02}$ |
| FARMI | 0.60 $\pm 0.07$ | 0.12 $\pm 0.03$ | 0.15 $\pm 0.02$ | 0.15 $\pm 0.02$ | 0.04 $\pm 0.01$ | 0.06 $\pm 0.03$ | FARMI | 0.35 $\pm 0.05$ | 0.11 $\pm 0.03$ | 0.09 $\pm 0.02$ | 0.05 $\pm 0.01$ |
| `FairCOCCO` | **0.49** $\pm \mathbf{0.06}$ | **0.08** $\pm \mathbf{0.02}$ | **0.05** $\pm \mathbf{0.01}$ | **0.07** $\pm \mathbf{0.02}$ | **0.03** $\pm \mathbf{0.01}$ | **0.04** $\pm \mathbf{0.01}$ | `FairCOCCO` | **0.33** $\pm \mathbf{0.02}$ | **0.06** $\pm \mathbf{0.02}$ | **0.03** $\pm \mathbf{0.01}$ | 0.04 $\pm 0.02$ |

**Calibration.** CAL requires conditional independence between target and sensitive attributes given predictions. As the conditioning variable is continuous, we report the `FairCOCCO` score on the same experiments. We see in Table 11 and 12 that `FairCOCCO` achieves superior fair and predictive outcomes under different definitions of fairness when compared to other methods.

Table 11: **Performance in binary setting.** Accuracy (ACC) and `FairCOCCO` (COCCO) under CAL; the best results are emboldened.

| Method | COMPAS | | German | | Drug | | Adult | |
|---|---|---|---|---|---|---|---|---|
| | ACC | COCCO | ACC | COCCO | ACC | COCCO | ACC | COCCO |
| Donini et al. (2018) | $0.76 \pm 0.03$ | $0.12 \pm 0.02$ | $0.70 \pm 0.05$ | $0.06 \pm 0.01$ | $0.80 \pm 0.07$ | $0.13 \pm 0.21$ | 0.78 | 0.16 |
| NN | $0.90 \pm 0.02$ | $0.07 \pm 0.02$ | $0.74 \pm 0.07$ | $0.07 \pm 0.03$ | $0.80 \pm 0.08$ | $0.24 \pm 0.08$ | 0.84 | 0.18 |
| Mary et al. (2019) | $0.87 \pm 0.12$ | $0.07 \pm 0.03$ | $0.71 \pm 0.11$ | $0.06 \pm 0.02$ | $\mathbf{0.79 \pm 0.03}$ | $\mathbf{0.08 \pm 0.03}$ | 0.81 | 0.15 |
| (Steinberg et al., 2020b) | $0.88 \pm 0.03$ | $0.06 \pm 0.01$ | $\mathbf{0.73 \pm 0.06}$ | $0.04 \pm 0.02$ | $0.77 \pm 0.05$ | $0.16 \pm 0.05$ | 0.80 | 0.14 |
| `FairCOCCO` | $\mathbf{0.89 \pm 0.02}$ | $\mathbf{0.02 \pm 0.02}$ | $0.71 \pm 0.05$ | $\mathbf{0.02 \pm 0.01}$ | $0.78 \pm 0.06$ | $0.11 \pm 0.06$ | **0.83** | **0.11** |

## C.2 FAIRCOCCO ESTIMATION

In this section, we provide additional results on convergence of `FairCOCCO Score` estimation as a function of batch size, similar to the experiment performed in the main paper. We show convergence on **Adult** and **German** dataset in Figure 2. We note that while convergence of estimation depends on properties of different datasets, the estimation of `FairCOCCO Score` stabilizes at batch sizes $> 256$.

---

[4] www.uniformguidelines.com.

Table 12: **Protection of multiple attributes**. Level of protection provided to individual attributes when all attributes are simultaneously protected under CAL. Lowest MSE and `FairCOCCO` score are emboldened. **(left)** C&C dataset, **(right)** Students dataset.

| Method | Joint MSE | Joint COCCO | racePctBlack COCCO | racePctWhite COCCO | racePctAsian COCCO | racePctHisp COCCO |
|---|---|---|---|---|---|---|
| *NN* | 0.22 ± 0.01 | 0.16 ± 0.03 | 0.16 ± 0.04 | 0.13 ± 0.08 | 0.07 ± 0.03 | 0.12 ± 0.03 |
| FACL | 0.55 ± 0.10 | 0.14 ± 0.02 | 0.11 ± 0.01 | 0.09 ± 0.03 | 0.11 ± 0.01 | 0.09 ± 0.04 |
| FARMI | 0.53 ± 0.05 | 0.15 ± 0.05 | 0.13 ± 0.02 | 0.12 ± 0.03 | 0.05 ± 0.01 | 0.10 ± 0.03 |
| FairCOCCO | **0.47 ± 0.09** | **0.06 ± 0.01** | **0.08 ± 0.01** | **0.07 ± 0.02** | **0.03 ± 0.01** | **0.06 ± 0.01** |

| Method | Joint MSE | Joint COCCO | age COCCO | sex COCCO |
|---|---|---|---|---|
| *NN* | 0.25 ± 0.05 | 0.11 ± 0.05 | 0.09 ± 0.01 | 0.05 ± 0.06 |
| FACL | 0.32 ± 0.03 | 0.14 ± 0.02 | 0.12 ± 0.02 | 0.07 ± 0.02 |
| FARMI | 0.36 ± 0.06 | 0.15 ± 0.01 | 0.11 ± 0.02 | 0.10 ± 0.02 |
| FairCOCCO | **0.37 ± 0.05** | **0.04 ± 0.02** | **0.06 ± 0.01** | **0.03 ± 0.03** |

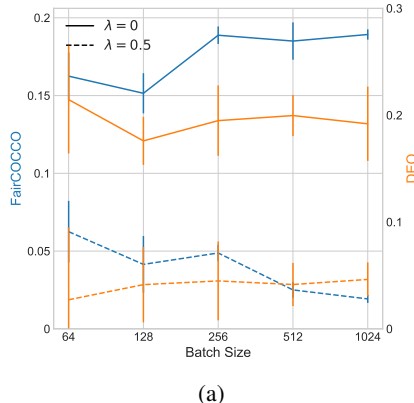 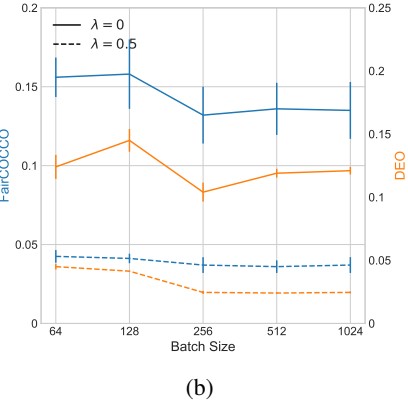

(a)        (b)

Figure 2: **Estimation of FairCOCCO Score.** **(a)** Adult dataset, **(b)** German dataset.

## C.3 STATISTICAL TESTING

We demonstrate how the proposed fairness measures can be employed as a test statistic to perform statistical tests, resulting in stronger guarantees and transparency (Fukumizu et al., 2007; Gretton et al., 2005). We highlight that while other fairness measures (MI and MCC) can be developed as test statistics, the empirical estimation of these measures involve multiple levels of approximations, and it is unclear whether the approximated statistics still retain the theoretical properties. Figure 3 shows the distributions of predictions with fairness regularization. Notably, EO only requires statistical independence between predictions and sensitive attributes given true outcome, whereas DP enforces "strict" independence between predictions and attributes.

Table 13: **Statistical testing.** Accuracy-fairness trade-offs under different fairness notions and corresponding test of statistical significance. **(left)** EO setting, **(right)** DP setting.

| $\lambda$ | ACC | DEO | COCCO | $p$-value |
|---|---|---|---|---|
| 0.0 | 78.33 | 0.66 | 0.21 | 0.00 |
| 0.2 | 76.67 | 0.39 | 0.14 | 0.14 |
| 0.5 | 70.36 | 0.07 | 0.03 | 0.45 |
| 1.0 | 67.78 | 0.03 | 0.02 | 0.74 |
| 2.0 | 60.57 | 0.00 | 0.01 | 0.90 |

| $\lambda$ | ACC | DI | COCCO | $p$-value |
|---|---|---|---|---|
| 0.0 | 78.33 | 3.05 | 0.07 | 0.00 |
| 0.2 | 72.56 | 1.54 | 0.03 | 0.04 |
| 0.5 | 69.33 | 1.77 | 0.01 | 0.09 |
| 1.0 | 67.38 | 1.13 | 0.01 | 0.14 |
| 2.0 | 64.60 | 0.92 | 0.00 | 0.27 |

As the null distribution is not known (Fukumizu et al., 2007), permutation testing is performed. Table 13 reveals the accuracy-fairness trade-offs and $p$-values under different regulation strengths. The $p$-values indicate the probability of observing the test statistic under null hypothesis of (conditional) independence. As we expect, stronger fairness regularization leads to lower levels of unfairness as measured by DI and DEO, as well as stronger guarantees in statistical tests. For example, at $\lambda = 2.0$, we can say with $90\%$ chance that predictions are conditionally independent of sensitive attributes (under EO) or $27\%$ chance that predictions are independent of sensitive attributes (under DP).

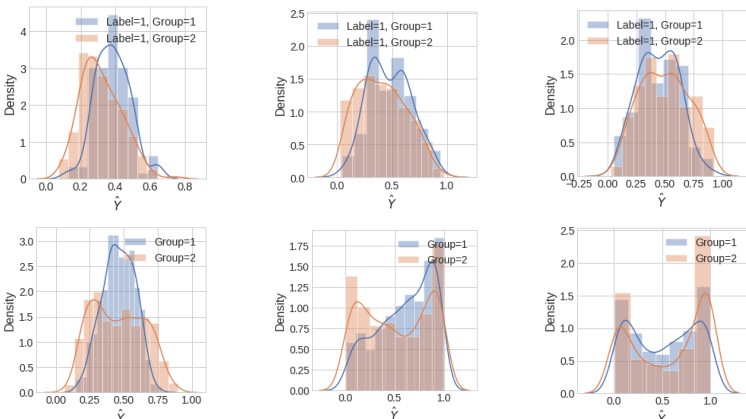

Figure 3: **Visualizing FairCOCCO regularization. (Top)** distribution of predictions for label 1 of different group memberships under EO. **(Bottom)** distribution of predictions for different group memberships under DP. Predictions are produced by regularized logistic regression model with $\lambda = 0, \lambda = 0.5, \lambda = 1.0$, respectively, across each row.

## C.4 SENSITIVITY ANALYSIS: ACCURACY-FAIRNESS TRADE-OFFS

One of the key contributions of this study is the introduction of a differentiable fairness penalty that can naturally extend to multiple sensitive attributes. In this section, we generate the frontier of possible values on three experiments to better evaluate the sensitivity of our proposed methods to different numbers of sensitive attributes:

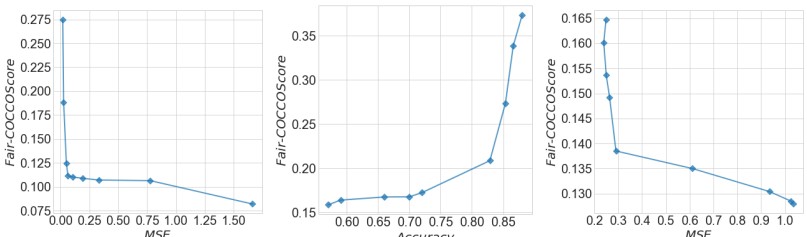

Figure 4: **Fairness-accuracy trade-off. (left)** C&C dataset with four sensitive attributes; **(middle)** students dataset with two sensitive attributes; **(right)** drugs dataset with three sensitive attributes.

- Regression on C&C with 4 attributes: `racePctBlack`, `racePctAsian`, `racePctWhite`, and `racePctHisp`,
- Regression on Students with 2 attributes: `age` and `gender`,
- Binary classification task on Drugs with 3 attributes: `age`, `gender`, and `ethnicity`.

As Figure 4 illustrates, similarly, fairness and prediction outcomes are achieved at various number of sensitive attributes.

