# OpenReview forum: "Practical Approaches for Fair Learning with Multitype and Multivariate Sensitive Attributes"
_ICLR.cc/2023/Conference — Submitted to ICLR 2023_

### Official Review · Reviewer_iTEE · 2022-10-21

**Confidence:** 5
**Correctness:** 2
**Technical Novelty And Significance:** 2
**Empirical Novelty And Significance:** 2
**Recommendation:** 3

**Clarity, Quality, Novelty And Reproducibility:**

The paper is somewhat clear, but some important details are missing or unclear. There are enough resources and details to reproduce the results. But the main ideas of the paper are not novel or have limited novelty. Furthermore, provided experimental analysis is not enough to support the claim.

**Strength And Weaknesses:**

The paper aims to solve an important problem which is ensuring fairness for any number and any type of protected attributes.

But the paper suffers from lots of weaknesses:
- The multiple protected attribute-based fairness notions are not new. There are several existing works on this as follows: Multicalibration (U. Hebert-Johnson et al, 2018), subgroup fairness (M. Kearns et al, 2018), and intersectional fairness (J. Foulds et al, 2020). The paper doesn't motivate enough why we need a new multiple attribute-based fairness metric?
- The proposed learning algorithm is also not new. Adding fairness metric as a regularization term with a trade-off parameter is the standard approach in many existing fair ML work. For example, Foulds et al, 2020 developed a backpropagation-based learning algorithm with the regularization term, designed using intersectional fairness criteria.
- Regarding the discrete and continuous outcomes. Agarwal et al 2018 proposed a reduction-based constrained optimization method that can ensure fairness in classification task for single or multiple protected attributes. Furthermore, they extended their reduction based method for regression task (Agarwal et al 2019) with fairness metric for continuous outcomes.
- I am very surprised that authors didn't compare/analyze their proposed metric and learning algorithms with any of the multiple protected attribute-based existing fair ML works in their experiments. This is the biggest weakness of the paper.



**Summary Of The Paper:**

The paper presents a new fairness metric that can measure fairness for multiple sensitive attributes of any type. The author also developed a learning algorithm by using their fairness criteria as a regularization term.

**Summary Of The Review:**

See the comments in the "weakness" section.

..........................................
I thank the authors for explaining some of the concerns. However, I still think that the novelty of the paper is limited and not enough for ICLR. However, after reading other reviewers comments and authors feedback, I've decided to increase my overall rating.

---

> ### Author Response · Authors · 2022-11-19
> **Response to Reviewer iTEE**
>
> Thank you very much for your helpful feedback and comments! We aim to address all the individual points in your review here.
>
> ---
> ## Q1 Related Works
>
> *There are several existing works on this as follows: Multicalibration (U. Hebert-Johnson et al, 2018), subgroup fairness (M. Kearns et al, 2018), and intersectional fairness (J. Foulds et al, 2020).*
>
> Thank you for pointing out related work. We would like to highlight that the key contribution of our method is in proposing a fairness metric (and regularizer) that is 1) compatible with all dependency-based notions of fairness (incl. DP, EO, and CAL), 2) supports settings with continuous/discrete outcomes and sensitive attributes, 3) multiple sensitive attributes, and 4) can capture higher-order fairness effects. We discuss each work in turn to higlight how the novelty differs from ours:
>
> * Multicalibration (U. Hebert-Johnson et al, 2018): this work aims to ensure classification predictions for subgroups are calibrated. Both subgroup membership and label are assumed to be **discrete** variables, and cannot be easily extended to **continuous** variables. Additionally, the method is only compatible with calibration, and not other popular definitions of group fairness, namely demographic parity, and equalized odds.
>
> * Subgroup fairness (M. Kearns et al, 2018): Similarly, this work focuses on settings where subgroup membership can be **categorically** identified, which cannot be achieved for continuous sensitive attributes (without introducing discretization effects, which itself can induce bias).
>
> * Intersectional fairness (J. Foulds et al, 2020): Intersectional fairness considers combinatorial combination of **discrete** attributes, but encounters challenges in effectively estimating fairness effects when encountering data sparsity issues in each intersectional bin. In addition, it is only applicable to **unconditional** definitions of fairness (specifically demographic parity).
>
> * Agarwal et al 2018: This work also only considers multiple **binary** sensitive attributes, without any discussion on how to address **continuous** ones. More importantly, it only captures first-order, **linear effects** of sensitive attributes on predictions. This can bias fairness estimations negatively in practical settings, where the relationship between sensitive attribute and outcome is likely highly nonlinear.
>
> * Agarwal et al 2019: This work is the only one that can assess fairness wrt continuous outcomes, but is still only applicable to **binary** sensitive attributes. Similar assumptions on linear fairness effects are likely to be detrimental when deployed in the real-world.
>
> While the identified works represent valuable methods of fairness inspection, the key focus of our work is in providing a practical tool that estimates fairness at a distribution-level, without examining subgroup outcomes in intersectional confusion matrices. We do not believe the related works take away from the novelty and contribution of ours. With that being said, we will include a discussion with the recommended related works in the camera-ready version of the paper (if accepted).
>
> ---
> ## Q2 Fair Learning
>
> *The proposed learning algorithm is also not new. Adding fairness metric as a regularization term with a trade-off parameter is the standard approach in many existing fair ML work.*
>
> We agree with your comment! In fact, all the baselines we compare against incorporate fairness constraints as a relaxed Lagrangian (or regularization). We *do not* claim that the framework of fairness regularization is a *new* contribution in our work. Rather, we wish to demonstrate that our method of fairness estimation can strongly capture dependence and thus lead to more optimal prediction-fairness trade-offs than existing regularization-based fairML works.
>
> ---
> ## Q3 Baseline Comparison
> *I am very surprised that authors didn't compare/analyze their proposed metric and learning algorithms with any of the multiple protected attribute-based existing fair ML works in their experiments.*
>
> As we highlighted in our response Q1, none of the suggested references can adequately address the fairness challenges we described. The most relevant work that we found is Mary et al. 2019, which supports single (but not multiple) continuous sensitive attributes and continuous outcome. In an extensive suite of experiments, we demonstrate that our method is superior to Mary et al 2019 (which we adapt to perform in the multivariate setting). As far as we know, our work is the first to address this major gap in algorithmic fairness literature.
>
> ---
>
> Thank you again for your help in improving our work! Please let us know if our latest changes have addressed your concerns.

---

> ### Author Response · Authors · 2022-11-25
> **Author Follow-up**
>
> Dear Reviewer iTEE,
>
> Once again, **thank you** for your thoughtful review. While we have this current period of discussion, please do let us know if our response has addressed your concerns - we are keen to keep engaging with you to address any additional questions or comments.
>
> Best wishes,
>
> The Authors

---

### Official Review · Reviewer_V8La · 2022-10-27

**Confidence:** 5
**Clarity, Quality, Novelty And Reproducibility:** The paper is written clearly.
**Correctness:** 4
**Technical Novelty And Significance:** 4
**Empirical Novelty And Significance:** 3
**Recommendation:** 5

**Strength And Weaknesses:**

Strength
- The paper proposed a novel in-processing based fairness-aware learning method based on FairCOCCO, which can be computed via using kernel based conditional cross covariance operator.
- The experiments show that the method can be applied in various setting that involve multi-variate, continuous-valued sensitive attributes.
- It outperforms several baselines on multiple settings.
- Detailed analyses are given in Supplementary.

Weakness
- Some recent baselines are missing, e.g.,
 [Jung et el.,  Learning fair classifiers with partially annotated group labels, CVPR 2021]
[Quadrianto et al., Discovering fair representations in the data domain. CVPR 2019]
[Jiang et al., Identifying and correcting label bias in machine learning. AISTATS 2020]
- Result on COMPAS is a bit dubious -- DEO is 0.00??
- While the result on the relaxation on computing COCCO score is given in the appendix, what is the actual computational complexity? Fairly comparing with the CPU time would be also beneficial.
- Some more results on vision dataset such as UTKFace would be also helpful.


**Summary Of The Paper:**

The paper proposes to practical method called FairCOCCO that can incorporate multitype/multivariation sensitive attributes for fairness-aware learning. The method simply develops the kernel-based measure of fairness and uses it as a regularizer for learning a predictive model. The experimental results seem promising, but the comparing baselines are relatively old.


**Summary Of The Review:**

Overall, while the proposed method is simple, the experimental results seem promising. The actual computational complexity comparison would be helpful for making the final decision.

---

> ### Author Response · Authors · 2022-11-19
> **Response to Reviewer V8La**
>
> Thank you very much for your helpful feedback and comments! We aim to address all the individual points in your review here.
>
> ---
> ## Q1 Baseline Comparison
>
> *Some recent baselines are missing, e.g., [Jung et el., Learning fair classifiers with partially annotated group labels, CVPR 2021] [Quadrianto et al., Discovering fair representations in the data domain. CVPR 2019] [Jiang et al., Identifying and correcting label bias in machine learning. AISTATS 2020]*
>
> Thank you for pointing out related work. We would like to highlight that the key contribution of our method is in proposing a fairness metric (and regularizer) that is 1) compatible with all dependency-based notions of fairness (incl. DP, EO, and CAL), 2) supports settings with continuous/discrete outcomes and sensitive attributes, 3) multiple sensitive attributes, and 4) can capture higher-order fairness effects. We discuss each work in turn to higlight how the novelty differs from ours:
>
> * Jung et al, 2021: This is a very interesting work that asks and addresses the question *what if we do not know the sensitive attribute (for example, due to privacy concerns)?* This is an important challenge, but is orthogonal to the aforementioned challenges that we address in this work.
>
> * Quadrianto et al, 2019: This work addresses fairML through *pre-processing*, aiming to learn fair representations that can be employed for any downstream task. The focus is different to ours, which is to introduce a fairness metric for settings with *multiple*, *arbitrary* sensitive attributes and where the label can be *continuous or discrete* and to introduce an *in-processing* regularization method that can be applied to any learning model.
>
> * Jiang et al, 2019: This work takes an interesting theoretical angle to identify unlabeled bias, and de-biases learning algorithm through data re-weighting. Additionally, we see that this work can be applied to non-gradient based learning algorithms, which is very valuable as decision-tree based methods are SOTA in tabular settings, where most fairML problems arise! However, we wish to re-iterate that our proposed method can quantify fairness (which this work cannot) and reduce it through regularization.
>
> We will include these discussions in the camera-ready version of the paper (if accepted). What we hope to emphasize is that our work focuses on addressing a different, but equally pertinent set of challenges. Therefore, the suggested works do not reduce our work's contributions.
>
> ---
>
> ## Q2 Results on COMPAS
>
> *Result on COMPAS is a bit dubious -- DEO is 0.00??*
>
> Thank you for this comment. To clarify, this is due to rounding down, the exact DEO is 0.004.
>
> ---
>
> ## Q3 Computation Time
>
> *While the result on the relaxation on computing COCCO score is given in the appendix, what is the actual computational complexity? Fairly comparing with the CPU time would be also beneficial.*
>
> Thank you for raising a great point. Indeed, our theoretical analysis indicated that computational complexity could be a major limitation of our work. To address this, we introduced two ways to relax complexity, namely through low-rank Cholesky decomposition and mini-batch estimation. Practically, we found that, there is no notable increase in CPU time when using our method for fair learning. For reference, on Adult (one of the largest datasets we consider), the unregularized NN takes approx $50$ minutes to complete $10$ runs, and the NN regularized with FairCOCCO took just shy of $60$ minutes.
>
> ---
>
> ## Q4 Additional benchmark
>
> *Some more results on vision dataset such as UTKFace would be also helpful.*
>
> Thank you for this suggestion. We are happy to include additional results on UTKFace for the camera-ready (if accepted). We hope that you can understand that it is challenging to run a large-scale image experiment for all baselines with appropriate tuning and stochastic runs given compute and time constraints.
>
> ---
>
> Thank you again for your help in improving our work! Please let us know if our latest changes have addressed your concerns.

---

> ### Author Response · Authors · 2022-11-25
> **Author Follow-up**
>
> Dear Reviewer V8La,
>
> Once again, **thank you** for your thoughtful review. While we have this current period of discussion, please do let us know if our response has addressed your concerns - we are keen to keep engaging with you to address any additional questions or comments.
>
> Best wishes,
>
> The Authors

---

### Official Review · Reviewer_yfN2 · 2022-11-02

**Confidence:** 4
**Correctness:** 4
**Technical Novelty And Significance:** 3
**Empirical Novelty And Significance:** 3
**Recommendation:** 6

**Clarity, Quality, Novelty And Reproducibility:**

The paper is well written, there is enough detail on the proposed method to reproduce the proposed method, though some details such as hyper-parameter tuning on regularization strength seem to be omitted.

The novelty of the paper is somewhat limited, since it seems to be a straightforward application of Fukumizu 2007 in the context of fairness. However, the proposition seems solid and interesting.

**Strength And Weaknesses:**

Strengths:

The paper does a good job on explaining the merits of the cross covariance operator as a viable tool to extend DP, EO and Calibration to multivariate and potentially continuous target and sensitive attributes. The presentation is solid, the regularization objective is well motivated, and the experimental results  show improvement over the examined baselines.

Weaknesses:

I think the novelty of the work is somewhat limited. The main theoretical results presented in the paper are taken directly from Fukumizu 2007, though the application to fairness metric is, to my knowledge, novel.

Some key implementation details are shown in the appendix, rather than discussed in the main paper, such as the closed form expression for FairCOCCO in Appendix A, which also seems taken from Fukumizu 2007.

Upon seeing the discussion on Appendix C.2 on the impact of batch size on the FairCOCCO measure, I would like to ask the authors if comparable hyper parameter optimization was also performed for the competing baseline methods. I perhaps missed a similar analysis on how important the regularization strength term $\lambda$ was to the final results.


**Summary Of The Paper:**

The paper addresses the extension of existing fairness notions in ML such as Demographic Parity, Equal Opportunity, and Calibration to settings where the both the target and sensitive attributes are potentially multivariate and continuous.

 It proposes a metric (FairCOCCO) based on the cross covariance operator over reproducing kernel Hilbert spaces that can be used to both measure and regularize existing ML systems for fairness-aware learning, and shows comparable or better results over existing methods on multiple real-world datasets.

**Summary Of The Review:**

I think the paper has merit, and proposes a well-motivated and grounded application of existing literature to easily extend existing fairness definitions into multivariate target and attribute scenarios. The main drawback of the paper seem to be the relatively equivalent results to existing methods, and a potential lack of technical novelty.

---

> ### Author Response · Authors · 2022-11-19
> **Response to Reviewer yfN2**
>
> Thank you very much for your helpful feedback and comments! We aim to address all the individual points in your review here.
>
> ---
> ## Q1 Novelty
> *I think the novelty of the work is somewhat limited. The main theoretical results presented in the paper are taken directly from Fukumizu 2007, though the application to fairness metric is, to my knowledge, novel.*
>
> We agree with you! It was our intention to fairly credit original works (incl Fukumizu, 2007), which provided a sound theoretical basis (especially around covariance operators and RKHS) on which we built our practical fairness method, and we never wanted to claim undeserved theoretical contributions. Please also see our discussions with **Reviewer V8La** and **Reviewer iTEE** on related works for more on the positioning of this work.
>
> We identified a major gap in algorithmic fairness, concerning the quantification and protection of fairness in problems with *multiple*, *arbitrary* sensitive attributes and *continuous/discrete* outcomes and identified cross-covariance operators as the most viable tool. Specifically, they can strongly capture fairness effects (without parametric assumptions, or first-order relaxations) while being algebraically tractable, and easy to incorporate into *autograd* frameworks that are crucial for DL systems. As you stated, we introduced a normalization scheme to adapt the cross-covariance operators into a useful metric and regularization method.
>
> What followed was extensive investigation into the empirical merits of our approach. Through a suite of real-world datasets and more complicated time-series and image tasks, we found that our practical tool consistently achieved a strong prediction-fairness tradeoff. We believe that this work presents an important toolbox for fairML practitioners and also sets a strong benchmark for future works.
>
> ---
> ## Q2 Hyperparameter Tuning
>
> *I would like to ask the authors if comparable hyper parameter optimization was also performed for the competing baseline methods. I perhaps missed a similar analysis on how important the regularization strength term was to the final results.*
>
> Thank you for this comment. Yes we do. We would like to point you to *Appendix B.1.1 Model Details*, where we discussed the hyperparameters we considered. Specifically, we tuned batch size $\in \{64, 128, 256\}$, learning rate $\in \{1e-2, 1e-3, 1e-4\}$, and fairness penalty $\lambda \in \{0.0, 0.5, 1.0, 2.0, 5.0\}$. We impose a **search budget** of 10 iterations using Bayesian Optimization, where the search objective was validation set loss. We believe these procedures guarantee a fair comparison.
>
>
> ---
> Thank you again for your help in improving our work! Please let us know if our latest changes have addressed your concerns.

---

> ### Author Response · Authors · 2022-11-25
> **Author Follow-up**
>
> Dear Reviewer yfN2,
>
> Once again, **thank you** for your thoughtful review. While we have this current period of discussion, please do let us know if our response has addressed your concerns - we are keen to keep engaging with you to address any additional questions or comments.
>
> Best wishes,
>
> The Authors

---

### Official Review · Reviewer_stjQ · 2022-11-03

**Confidence:** 3
**Clarity, Quality, Novelty And Reproducibility:** 1. On clarity
**Correctness:** 3
**Technical Novelty And Significance:** 2
**Empirical Novelty And Significance:** 3
**Recommendation:** 5

**Strength And Weaknesses:**

Strength:
1. Unlike most existing studies that focused on measuring/remedying unfairness against a single binary sensitive attribute, this paper focuses on the more refined (sub-)groups that are potentially distinguished based on multiple, continuous sensitive attributes
2. The paper is well-organized, and the literature review seems to be extensive.
3. The paper conducted extensive experiments on multiple datasets and compared the performance of the proposed method with many existing algorithms.

Weakness:
1. The paper proposes a new kernel method-based fairness measure. However, there is no theoretical support to justify why the proposed measure is better than the existing measures. Its performance is only evaluated empirically in experiments.
3. On the comparison in experiments: the authors compared the proposed FairCOCCO with other methods on various datasets. Many results are competitive, and it is not obvious that FairCOCCO outperforms other methods. For example, in Table 9, the bolded results are not the best, and the accuracy and fairness are very similar under different algorithms.

**Summary Of The Paper:**

The paper proposes a new fairness metric, FairCOCCO SCORE, that can quantify unfairness among groups are distinguished based on multiple sensitive attributes of different types. It measures the unfairness in reproducing kernel Hilbert space based on cross-covariance operators. Based on this measure, the paper further proposes a regularization term that can be used for learning fair predictors.


**Summary Of The Review:**

While the paper proposes a new fairness measure that is applicable to settings where groups are distinguished by (continuous) multiple sensitive attributes, it is not clear to me why the proposed measure is better than the existing unfairness measures that are also applicable to continuous sensitive attributes.

---

> ### Author Response · Authors · 2022-11-19
> **Response to Reviewer stjQ**
>
> Thank you very much for your helpful feedback and comments! We aim to address all the individual points in your review here.
>
> ---
> ## Q1 Theoretical Justification
>
> *However, there is no theoretical support to justify why the proposed measure is better than the existing measures. Its performance is only evaluated empirically in experiments.*
>
> Thank you for this comment. We did not provide much theoretical analysis around the merits of our approach, as we build on a comprehensive foundation of literature on cross-covariance operators and RKHS measures (see [1-4] for a few). Much of group fairness, including DP, EO, and CAL, can be boiled down to some permutation of dependence measurement between sensitive attributes, outcomes, and labels. While often unnecessarily obscured in existing literature, fairness quantification is often independent of the type (continuous or discrete), dimensionality of variables considered and fairness notion, and can be *generalized* to a dependence estimation task.
>
> Based on this insight, we identified cross-covariance operators as strong conditional dependence estimators with appealing theoretical guarantees. However, its use in fairML, especially for conditional fairness notions and multivariate problems, has not been explored. Our goal was to introduce necessary modifications (for example, to extend the measure to a normalized metric) but more importantly to empirically validate the merits of this approach. We conducted extensive experiments, especially around multivariate sensitive attributes (which existing methods cannot address), and found that our method consistently outperformed baseline methods. As such, we believe it is an important practical tool for fairML.
>
> Please let us know if you would like to see additional theoretical support.
>
>
> **References**
>
> [1] Bach, F.R. and Jordan, M.I., 2002. Kernel independent component analysis. Journal of machine learning research, 3(Jul), pp.1-48.
>
> [2] Gretton, A., Bousquet, O., Smola, A. and Schölkopf, B., 2005, October. Measuring statistical dependence with Hilbert-Schmidt norms. In International conference on algorithmic learning theory (pp. 63-77). Springer, Berlin, Heidelberg.
>
> [3] Fukumizu, K., Gretton, A., Sun, X. and Schölkopf, B., 2007. Kernel measures of conditional dependence. Advances in neural information processing systems, 20.
>
> [4] Baker, C.R., 1973. Joint measures and cross-covariance operators. Transactions of the American Mathematical Society, 186, pp.273-289.
>
> ---
> ## Q2 Performance
>
> *Many results are competitive, and it is not obvious that FairCOCCO outperforms other methods.*
>
> Indeed, if we solely look at FairCOCCO performance in the binary setting (reported in Table 3, 9, 10, where the problem can be reduced to the much simpler rates of outcome in a confusion matrix), it is not obvious that FairCOCCO outperforms baselines. In this simpler problem setting, most methods are comparable, and our aim is more to prove that our method is competitive with SOTA methods on a standardized benchmark. As we stated in *Section 4.1*:
>
> *'While the focus of this work is on introducing practical methods for fairness in multitype, multivariate settings, we want to first prove that FairCOCCO is also competitive with state-of-the-art methods on problems with binary sensitive attributes and outcomes.'*
>
> What we really wanted to highlight is the performance of our method in protecting fairness when there are multiple sensitive attributes and outcomes of arbitrary type. This is an area in which, as far as we know, no work has addressed. Through extensive experiments, we demonstrated that our method can protect joint and individual fairness, setting a strong benchmark that we hope future works can improve on.
>
> ---
> Thank you again for your help in improving our work! Please let us know if our latest changes have addressed your concerns.

---

> ### Author Response · Authors · 2022-11-25
> **Author Follow-up**
>
> Dear Reviewer stjQ,
>
> Once again, **thank you** for your thoughtful review. While we have this current period of discussion, please do let us know if our response has addressed your concerns - we are keen to keep engaging with you to address any additional questions or comments.
>
> Best wishes,
>
> The Authors

---

### Decision · Program_Chairs · 2023-01-20

**Decision:**

Reject

**Justification For Why Not Higher Score:**

Lack of novelty, insufficient motivation, weak experimental results.

**Justification For Why Not Lower Score:**

N/A

**Metareview: Summary, Strengths And Weaknesses:**

The paper introduces a fairness metric based on the cross covariance operator over reproducing kernel Hilbert spaces that can be used to both measure and regularize existing ML systems for fairness-aware learning. The reviewers recognized the work has some merit, however, they questioned the novelty of the proposed approach and they highlighted that both the motivation and the experimental evaluation could be stronger. I encourage the authors to take into account their reviews to revise their paper and resubmit to another top tier venue.